# Quantum Spectral Operator Learning for Solving Partial Differential Equations

## Abstract

Partial differential equations (PDEs) are central to modeling physical and engineering systems. Operator learning approximates their solution operators, enabling fast inference after training across diverse problem instances and strong generalization. While recent advances have proposed unsupervised methods that mitigate the cost of data generation, classical neural network–based approaches remain computationally expensive for high-dimensional operators and fine-resolution problems. To address these challenges, we propose a quantum–classical hybrid framework for unsupervised spectral operator learning. Our approach predicts spectral coefficients using quantum circuits, with gate parameters mapped from PDE instances (e.g., forcing functions or PDE parameters) via a classical neural network. To improve efficiency and feasibility, we introduce a training objective that requires fewer measurement repetitions than standard variational quantum linear solvers (VQLS). With this, we design shallower circuits by replacing controlled-unitary gates with direct Pauli measurements, which in turn allows grouping of commuting measurement operators for further reduction in runtime. The objective also resolves the sign ambiguity inherent in standard VQLS and guarantees recovery of the correct solution sign for PDEs. Overall, our framework reduces the computational cost and improves solution accuracy of VQLS, while also demonstrating the potential efficiency and scalability advantages of quantum operator learning over classical machine learning approaches. We validate our framework on one- and two-dimensional reaction–diffusion, Helmholtz, and convection–diffusion equations under diverse boundary conditions, achieving relative errors below 1%.

## 1 Introduction

Solving partial differential equations (PDEs) lies at the heart of scientific computing, underpinning advances in physics, engineering, and data-driven modeling. Despite decades of progress in numerical analysis, solving complex PDEs remains computationally prohibitive, particularly in high-dimensional domains or when fine spatial and temporal resolution is required. Traditional solvers, while accurate, often incur costs that scale poorly with the problem size, motivating the exploration of new algorithmic paradigms. While machine learning (ML) methods have been actively investigated to address these challenges (Li et al., 2020; Lu et al., 2021b;a), quantum computing (QC) has recently emerged as a promising alternative for improved efficiency and scalability (Kyriienko et al., 2021; Ali & Kabel, 2023; Morales et al., 2025).

Among the prominent directions in ML-based PDE solving is operator learning, where the goal is to approximate the mapping from PDE inputs—such as forcing terms, coefficients, or boundary conditions—to the corresponding solutions. By directly learning this operator, models such as the Fourier Neural Operator (FNO) (Li et al., 2020) and DeepONet (Lu et al., 2021a) enable fast inference, thereby significantly reducing the computational cost of solving PDEs repeatedly in parametric or multi-query settings. In parallel, quantum counterparts of operator networks have been proposed, such as quantum Fourier networks and

quantum DeepONet variants (Jain et al., 2024; Xiao et al., 2025). These operator networks have been applied to diverse scientific problems, from fluid dynamics to material science, highlighting their potential as surrogates for expensive numerical solvers (Azizzadenesheli et al., 2024). However, the utility of operator networks comes with an important limitation: they require large training datasets consisting of pre-computed PDE solutions. Generating such labeled data typically demands repeated runs of high-resolution numerical solvers, which can be prohibitively expensive, especially for nonlinear, multi-dimensional, or multi-physics PDEs. This data bottleneck has motivated unsupervised operator networks, which bypass labeled training data by embedding PDE structure into the learning objective (Li et al., 2021), including spectral operator learning that minimizes residuals in coefficient space (Choi et al., 2023; 2024). Although these methods eliminate the need for precomputation, classical algorithms based on neural networks still suffer from high training costs, particularly for high-dimensional operators or large basis expansions at high resolution.

To overcome these limitations, we propose a quantum-classical hybrid framework for unsupervised spectral operator learning. Our method combines the flexibility of neural operator learning with the efficiency of quantum linear solvers. Specifically, a classical neural network maps PDE inputs (e.g. forcing functions, PDE parameters, boundary conditions) to the parameters of a variational quantum circuit, while a quantum subroutine based on the Variational Quantum Linear Solver (VQLS) (Bravo-Prieto et al., 2023) prepares quantum states encoding spectral coefficients of the PDE solution. To enhance efficiency of VQLS, we design a novel training loss based on overlap of two quantum states, equipped with commutativity-aware Pauli measurement grouping. Compared to existing VQLS methods, it reduces the number of required Pauli observables to be measured from $O(K^2(\log K)^2)$ to $O(K \log K)$, where $K = N^d$, thus substantially enhancing training scalability with growing system size $N$ and spatial dimension $d$.

We validate our framework on a range of benchmark PDEs, including one- and two-dimensional reaction–diffusion, Helmholtz, and convection–diffusion equations with Dirichlet and Neumann boundary conditions. Across all experiments, the method achieves relative errors below 1% while demonstrating stable convergence. By eliminating the data-generation bottleneck and enabling generalization across PDE families without re-optimization, our framework offers a practical pathway for integrating quantum devices into PDE operator learning.

## 2 Preliminaries and Related Works

**Spectral Operator Learning**   We start by considering the reaction–diffusion equation defined on a bounded domain $\Omega \subset \mathbb{R}^n$, subject to boundary conditions, with a PDE parameter $\epsilon > 0$ and an external forcing term $f$:

$$
\begin{aligned}
-\epsilon \Delta u + u &= f, & x \in \Omega \\
\mathcal{B}(u) &= 0, & x \in \partial\Omega
\end{aligned}
\tag{1}
$$

where $\mathcal{B}$ represents a boundary operator, including Dirichlet, Neumann, and mixed boundary conditions. This example corresponds to the class of second-order elliptic PDEs, which also includes important models such as the Helmholtz and convection–diffusion equations. Thus, while we illustrate our method on the reaction–diffusion case for clarity, the formulation remains fully general. For the one-dimensional domain $I = [-1, 1] \subset \mathbb{R}$, the Legendre–Galerkin method provides an approximation to the solution $u$ in the form of $\hat{u}$:

$$
\hat{u}(x) = \sum_{k=0}^{N-2} \hat{\alpha}_k \phi_k(x),
\tag{2}
$$

where the basis functions $\phi_k = L_k + a_k L_{k+1} + b_k L_{k+2}$ are given by a compact linear combination of Legendre polynomials $\{L_k\}$, and $\hat{\alpha} := (\hat{\alpha}_0, \ldots, \hat{\alpha}_{N-2})$ represents the vector of their coefficients. The coefficients $a_k$ and $b_k$ are chosen to strongly enforce the exact boundary conditions, including Dirichlet, Neumann, and mixed types. Inspired by the weak formulation, the unsupervised Legendre–Galerkin neural network (ULGNet) proposed by (Choi

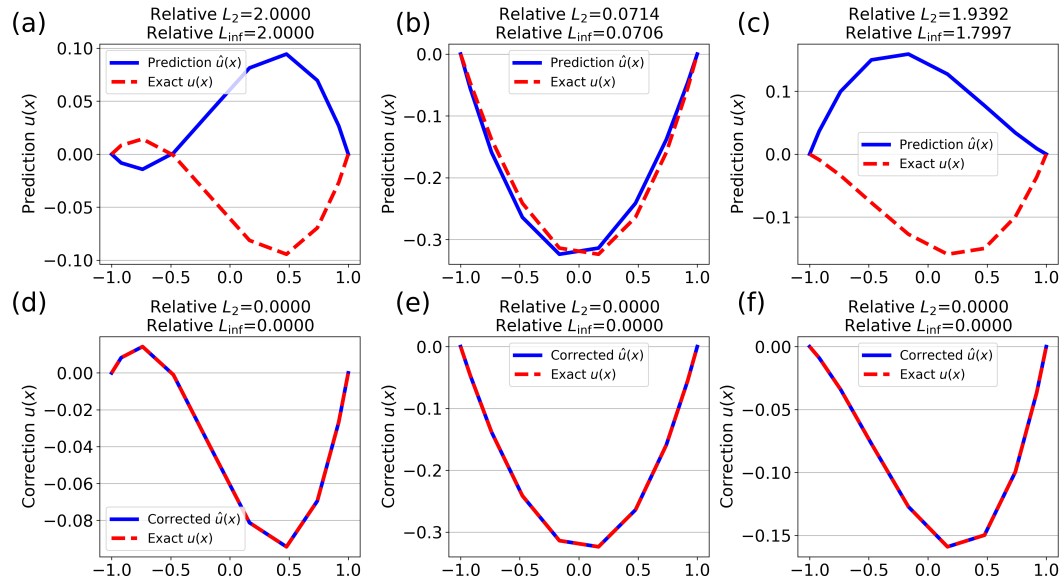

Figure 1: Illustration of the sign ambiguity in VQLS. Top row: Original VQLS predictions. Bottom row: Sign-corrected predictions. Left column: Reflection along the x-axis. Middle column: Reflection along the y-axis. Right column: Reflection along both axes.

et al., 2023; 2024) trains a neural network, parameterized by weights $w$, to predict the spectral coefficients $\hat{\alpha}$ of the approximate solution $\hat{u}$ from multiple forcing function inputs $f^{(i)}$, $i = 1, \ldots, D$, using the following loss:

$$
\mathcal{L}_{\text{ULGNet}}(w) = \sum_{i=1}^{D} \sum_{k=0}^{N-2} \left| -\epsilon \int_I \hat{u}_{xx}^{(i)} \, \phi_k \, dx + \int_I \hat{u}^{(i)} \, \phi_k \, dx - \int_I f^{(i)} \, \phi_k \, dx \right|^2
$$

$$
= \sum_{i=1}^{D} \| (-\epsilon S + M)\hat{\alpha}(f^{(i)}; w) - F^{(i)} \|_2^2
$$

(3)

where $F^{(i)}$ is the forward transform vector of the $i$-th forcing function and $\| \cdot \|_2$ represent the $L_2$ norm. The spectral matrix $A = -\epsilon S + M$ is composed of the stiffness matrix $S$ and the mass matrix $M$, which are diagonal and symmetric penta-diagonal, respectively. A complete derivation of the weak formulation together with the exact entries of the matrices S and M is presented in Appendix A.2. More generally, the size of the spectral matrix is $K = (N - 1)^d$, growing exponentially with $d$ and reflecting the curse of dimensionality inherent in spectral methods. Unlike physics-informed neural networks (PINNs) (Raissi et al., 2019) that typically learn a single instance, ULGNet leverages the residual loss to learn and predict across multiple instances. By leveraging the basis functions of spectral methods, the predicted solution exactly satisfies various boundary conditions. In general, this framework enables the neural network to take not only external forcing functions, but also PDE parameters and boundary conditions as inputs.

**VQLS** Given a linear system $Ax = b$, where $x \in \mathbb{R}^K$ is the solution vector, the variational quantum linear solver (VQLS) (Bravo-Prieto et al., 2023) is one of the most widely used approaches to solve such systems using variational quantum algorithms. In this framework, the matrix $A \in \mathbb{R}^{K \times K}$ is expressed, via a Pauli decomposition, as a linear combination of Pauli operators:

$$
A = \sum_{l}^{L} c_l A_l,
$$

(4)

where $c_l$ denotes the coefficient corresponding to the Pauli basis element $A_l$. The objective of the quantum linear system problem is to determine a quantum state $|x\rangle$ such that the

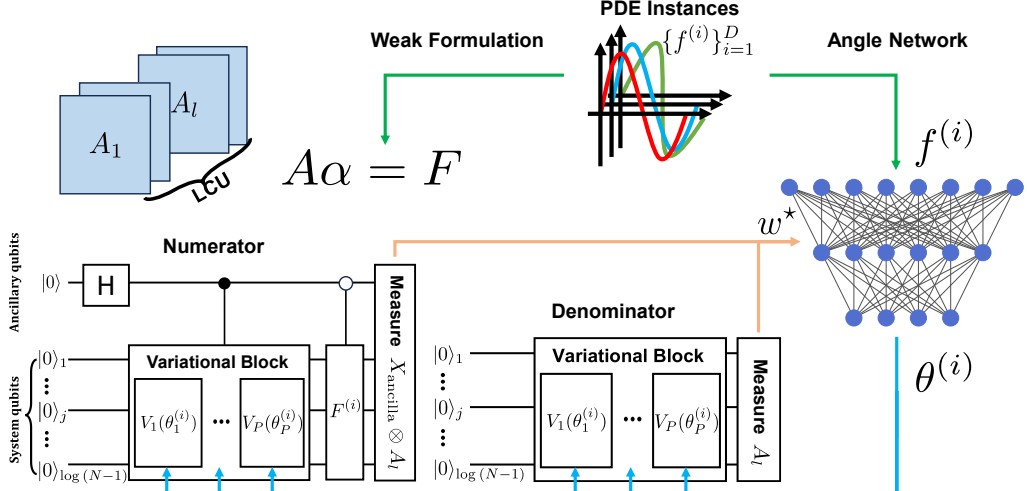

Figure 2: Overview of the proposed NVQLS framework. A classical network $g$ maps the input instance $f$ to circuit parameters $\theta = g(f; w)$; the parameterized quantum circuit $V(\theta)$ prepares $|\alpha\rangle$. Training minimizes the overlap loss with $A = \sum_l c_l A_l$. The numerator and denominator are estimated via Pauli measurements with commuting-term grouping to reduce shot complexity. After convergence, we rescale $\hat{\alpha} = (\|F\|\rangle/\|A|\hat{\alpha}\rangle\|) |\hat{\alpha}\rangle$ and reconstruct the solution $\hat{u}$ from $\hat{\alpha}$.

normalized vector $A|x\rangle/\|A|x\rangle\|_2$ is aligned with the normalized target state $|b\rangle$. Here, the quantum state is represented as $|x\rangle = V(\theta)|0\rangle$, where $V(\theta)$ denotes a parameterized quantum circuit. The loss function of VQLS is defined in terms of the squared magnitude of the inner product (i.e., the fidelity) as

$$\mathcal{L}_{\text{VQLS(global)}}(\theta) = 1 - \left|\frac{\langle b|A|x\rangle}{\|A|x\rangle\|_2}\right|^2 = 1 - \frac{\sum_{l,l'}^{L} c_{l'}^* c_l \langle 0|V(\theta)A_{l'}^\dagger|b\rangle\langle b|A_l|V(\theta)|0\rangle}{\sum_{l,l'}^{L} c_{l'}^* c_l \langle 0|V(\theta)|A_{l'}^\dagger A_l|V(\theta)|0\rangle} . \quad (5)$$

Due to the double summation over $l$ and $l'$, the number of Pauli measurements required via the Hadamard test becomes substantial, scaling as $O(L^2)$. Moreover, since it relies on a fidelity loss based on the squared magnitude, the standard VQLS framework ignores the relative phase and thus discards the solution's sign information, potentially causing issues in PDE applications.

**Related Works** Many studies have applied the VQLS framework to PDEs, but most efforts have been limited to relatively simple cases such as the Poisson and heat equations (Liu et al., 2021; 2022; Trahan et al., 2023; Ali & Kabel, 2023). Other works have focused on improving the practicality of VQLS and broadening its scope of applications (Turati et al., 2024; Pellow-Jarman et al., 2023; Patil et al., 2022; Pellow-Jarman et al., 2021; Surana & Gnanasekaran, 2024; Hosaka et al., 2023; Gnanasekaran & Surana, 2024). Moreover, there has been no attempt to generalize PDE solutions in a multi-instance setting using quantum variational algorithms, rather than being limited to a single forcing term.

## 3 Methodology

### 3.1 Model Architecture

Building on VQLS frameworks, we introduce the neural variational quantum linear solver (NVQLS), a hybrid variational quantum algorithm. NVQLS employ a classical neural network to predict spectral coefficients by mapping multiple data instances to circuit parameters and optimizing them via residual loss minimization. We further modify the loss structure and implementation scheme of VQLS, improving computational efficiency and mitigate the problem of local minima. Figure 2 illustrates an overview of our framework.

We simplify the formulation by defining $A = -\epsilon S + M \in \mathbb{R}^{(N-1)\times(N-1)}$ in Equation 3, yielding the linear system $A\alpha = F$, where $F$ is the forward transform of the forcing $f$. This system is solved by our hybrid quantum model NVQLS (denoted $h$), which maps $f \in \mathbb{R}^{N+1}$ to $\hat{\alpha} = h(f) \in \mathbb{R}^{N-1}$, then used in Equation 2 to construct the approximate solution $\hat{u}$. Inspired by VQLS, we employ a parameterized circuit $V(\theta)$ to prepare $|\hat{\alpha}\rangle = V(\theta)|0\rangle$, with $\theta$ optimized as a function of $f$ subject to $A|\hat{\alpha}\rangle/\|A|\hat{\alpha}\rangle\|_2 = F/\|F\|_2$. In general, for dimension $d$, the number of qubits $n$ is determined by $2^n = (N-1)^d$, where $N$ denotes the number of basis functions (or collocation points).

**Neural Embedding for Multi-instance Training**    In our method, the circuit parameters $\theta$ of $V(\theta)$ are produced by an angle network $g$, a classical neural network parameterized by trainable weights $w$, which maps the input forcing function $f$ to $\theta = g(f; w)$. Consequently, the variational circuit can be written as

$$V(\theta) = V(g(f; w)). \tag{6}$$

By leveraging the expressive power of the classical neural network, our method efficiently finds the optimal circuit paramters $\theta$ so that the model can generalize effectively across multiple inputs $\{f^{(i)}\}_{i=1}^{D}$. As the ansatz $V(\theta)$, we used a strongly entangling circuit (Schuld et al., 2020) and a hardware-efficient RY ansatz (Kandala et al., 2017), with parameters scaling in the number of qubits. The classical network $g$ employed fully connected layers. Further architectural details are given in Appendix A.3.

**Overlap Cost Function**    Unlike the original VQLS, which uses the squared magnitude of the inner product (i.e., fidelity) as in Equation 5, our method directly employs the real part of the inner product. We call this the overlap cost function. The resulting refined loss function is then defined as:

$$\mathcal{L}_{\text{NVQLS}}(w) = \frac{1}{D}\sum_{i=1}^{D}\left(1 - \frac{\sum_{l=1}^{L}\text{Re}\left(\langle F^{(i)}|A_l|\hat{\alpha}(f^{(i)}; w)\rangle\right)}{\sqrt{\sum_{l,l'=1}^{L}\langle\hat{\alpha}(f^{(i)}; w)|A_{l'}^{\dagger}A_l|\hat{\alpha}(f^{(i)}; w)\rangle}}\right), \tag{7}$$

where $D$ represents the number of data instances $f^{(i)}$, and $|\hat{\alpha}(f^{(i)}; w)\rangle = V(g(f^{(i)}; w))|0\rangle$ is expressed in a simplified notation. This overlap cost function replaces one of double summations in the original VQLS with a single summation over the real part of expectation values, significantly reducing the number of required circuit measurements and improving computational efficiency. Moreover, unlike the original VQLS loss, which discards the sign information of solutions, the overlap loss captures the relative phase between two quantum states, making it physically meaningful in various PDE settings. For training stability, we also utilize $\hat{\mathcal{L}}_{\text{NVQLS}}(w)$ which is obtained by multiplying the denominator and squaring with details in Appendix A.3.

**Implementation Scheme**    Unlike the original VQLS implementation, which employs the Hadamard test for both the numerator and the radicand in the denominator in Equation 5, our contribution lies in a novel method for implementing the cost function. This approach substantially reduces the computational overhead involved in cost evaluation, leading to more efficient quantum computations. In our cost function, Equation 7 (or Equation 33), two quantities need to be computed:

$$\beta_l^{(i)} \equiv \text{Re}\left(\langle F^{(i)}|A_l|\hat{\alpha}(f^{(i)}; w)\rangle\right), \qquad \gamma_{ll'}^{(i)} \equiv \langle\hat{\alpha}(f^{(i)}; w)|A_{l'}^{\dagger}A_l|\hat{\alpha}(f^{(i)}; w)\rangle. \tag{8}$$

where $A = \sum_{l=1}^{L} c_l A_l$, yielding $L$ distinct values for $\beta_l^{(i)}$ and $L(L+1)/2$ distinct values for $\gamma_{ll'}^{(i)}$ for each $F^{(i)}$. The key idea of our method is to estimate $\beta_l^{(i)}$ and $\gamma_{ll'}^{(i)}$ via Pauli measurements as illustrated in Figure 2. Moreover, for both $\beta^{(i)} \equiv \sum_l \beta_l^{(i)}$ and $\gamma^{(i)} \equiv \sum_{ll'} \gamma_{ll'}^{(i)}$ evaluations, the total number of measurements can be reduced by exploiting commutativity among observables, as the outcome of one observable can be reused to evaluate the expectation values of other commuting terms. To construct $\beta^{(i)}$, we group mutually commuting terms among the $L$ operators $A_l$ and, from each group, select one Pauli operator as the measurement observable. Using the measurement outcomes of the selected observables, the remaining terms within each group can be reconstructed to evaluate $\beta^{(i)}$. For $\gamma^{(i)}$, we first express each product $A_{l'}^{\dagger}A_l$ as a single Pauli string, thereby reducing the total number of

Table 1: Training Computational complexity of as a function of the matrix size $K = (N-1)^d$ where $N$ is the number of basis and $d$ represents the dimension of PDEs.

|  | ULGNet | VQLS | NVQLS (Ours) |
|---|---|---|---|
| Complexity measure | Arithmetic operations | Pauli measurements | Both measures |
| Neural embedding | $O(K^2)$ | - | $O(K \log K)$ |
| Loss evaluation | $O(K)$ | $O(K^2 (\log K)^2)$ | $O(K \log K)$ |
| Total complexity | $O(K^2)$ | $O(K^2 (\log K)^2)$ | $O(K \log K)$ |

distinct terms from $L(L+1)/2$ to $M$. Similar to the case of $\beta^{(i)}$, further grouping of commuting terms among the $M$ operators allows us to reduce the number of required measurements and to reconstruct $\sum_{l,l'} \gamma^{(i)}$. A detailed explanation of how commutativity reduces the cost of loss evaluation, along with a concrete example, is provided in Appendix A.5. Finally, the predicted vector $\hat{\alpha}$ of the predicted solution $\hat{u}$ is obtained from the optimized parameter $w^\star$ as

$$\hat{\alpha}(f; w^\star) = \frac{\|F\| |\hat{\alpha}(f; w^\star)\rangle}{\|A |\hat{\alpha}(f; w^\star)\rangle\|}. \tag{9}$$

## 3.2 Complexity Analysis

In this section, we analyze the computational complexity of the training procedure with respect to the matrix size $K = (N - 1)^d$, where $N$ denotes the number of basis functions and $d$ is the spatial dimension of the PDE. We analyze, in terms of big–$O$ notation, the costs of neural network embedding and loss evaluation for NVQLS, and compare them with those of classical ULGNet and the original VQLS. Table 1 provides a summary of the results.

For the neural network embedding, the angle network of NVQLS maps a $K$-dimensional input $f$ to the circuit parameters $\theta$ of size $O(n) = O(\log K)$, as the shallow ansatz $V(\theta)$ is utilized. It results in a basic arithmetic operation complexity of $O(K \log K)$ for computing $\theta$ via the angle network. Evaluating the loss requires Pauli measurements to compute $\beta$ and $\gamma$ in Equation 8, which correspond to the numerator and denominator terms in the NVQLS loss expression $1 - \beta/\gamma$. Assisted by grouping commuting Pauli operators, $\beta$ scales as $O(K)$ and $\gamma$ as $O(K \log K)$, increasing relatively slowly with $K$, as empirically shown in Figures 10 and 11 across all PDEs considered—reaction–diffusion, Helmholtz, and convection–diffusion. Combined with the neural embedding cost, this results in a total Pauli measurement complexity of $O(K \log K)$ for NVQLS.

On the other hand, the neural network of ULGNet directly outputs spectral coefficients of dimension $K$. Due to the matrix multiplication involved in the fully connected layer, this results in a computational complexity of $O(K^2)$. Although the loss evaluation in ULGNet scales as $O(K)$ thanks to the sparsity of the spectral matrix and assuming an efficient implementation, the primary bottleneck remains the network that directly produces the spectral coefficients. In VQLS, the number of Pauli measurements required to evaluate the loss scales as $O(L^2)$, where $L$ is the number of Pauli operators (Section 2). Using the empirical relation $L = O(K \log K)$ illustrated in Figure 12, the total cost in terms of Pauli measurements becomes $O(K^2 (\log K)^2)$.

Overall, the total computational complexity of NVQLS, $O(K \log K)$, demonstrates a clear advantage over the original VQLS, which scales as $O(K^2 (\log K)^2)$, and the classical ULGNet approach, with $O(K^2)$ complexity, highlighting the potential for quantum advantage. Our complexity analysis has so far focused on to the number of Pauli measurements, without considering the depth of quantum circuits. Once the complexity of quantum gate operations is taken into account, the advantage over VQLS becomes even more pronounced, since NVQLS requires a shallower circuit depth than VQLS. Although Big-O analysis may not capture the real computational overhead due to large hidden constants (e.g. repetitions for estimating the expectation value), the comparison between classical and quantum complex-

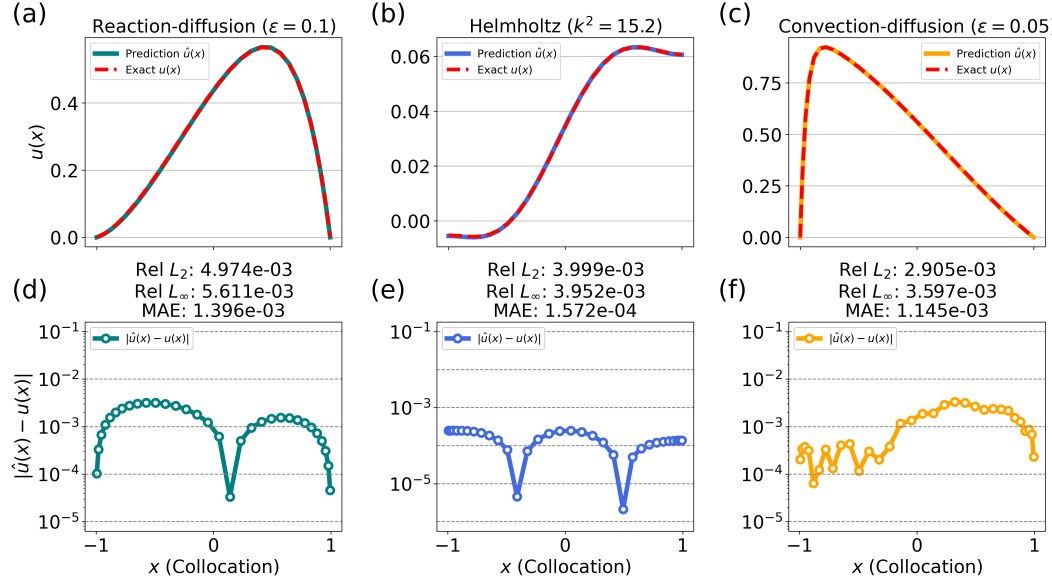

Figure 3: Numerical results. Top row: predicted solution $\hat{u}$ versus the exact solution $u$. Bottom row: absolute error $|\hat{u} - u|$. Left column: reaction–diffusion with $\epsilon = 0.1$ (Dirichlet BC). Middle column: Helmholtz with $k = 15.2$ (Neumann BC). Right column: convection–diffusion with $\epsilon = 0.05$ (Dirichlet BC).

ity measures highlights the potential for a quantum advantage. For further efficiency, we also analyze a truncation method detailed in Appendix A.9.

## 4 EXPERIMENT RESULTS

### 4.1 SPECTRAL OPERATOR LEARNING WITH FORCING INSTANCES

We describe the experimental setup and numerical results for various one-dimensional equations to assess the numerical accuracy of our model. In these experiments, the training objective of NVLQS is to map a set of forcing functions, serving as model inputs, to the corresponding spectral coefficients of the solutions. To this end, we generate input forcings $f^{(i)}$ $(i = 1, 2, \cdots, D)$ as random sums of trigonometric functions of the form

$$f^{(i)}(x) = h_1^{(i)} \sin(m_1^{(i)} x) + h_2^{(i)} \cos(m_2^{(i)} x) \tag{10}$$

where the coefficients $h_1^{(i)}, h_2^{(i)}, m_1^{(i)}, m_2^{(i)}$ are drawn from the uniform distribution on $[0, 1]$. The functions and their corresponding predicted solutions are then evaluated at collocation points $x_{k\,k=0}^{N}$, whose number matches the number of basis functions. Figure 3 summarizes the numerical results. Figures 3(a)–(c) present the solutions $u$ and predictions $\hat{u}$ for the reaction–diffusion, Helmholtz, and convection–diffusion equations, with representative parameter values indicated in the figures. The experiments mathematically enforce both Dirichlet and Neumann boundary conditions, where Dirichlet conditions are realized via basis functions

$$\phi_k(x) = L_k(x) - L_{k+2}(x), \tag{11}$$

while a different set is adopted for Neumann conditions, as specified in Equation 48. These experiments on various boundary conditions highlight the flexibility and robustness of NVLQS in accurately predicting solutions across different PDE constraints. Figures 3(d)–(f) show the absolute error $|\hat{u} - u|$ along with the corresponding error metrics for each equation. The relative $L_2$ and $L_\infty$ errors remain below 1%, and the mean absolute error (MAE) is approximately 0.1% in all cases, demonstrating the strong generalization capability of the

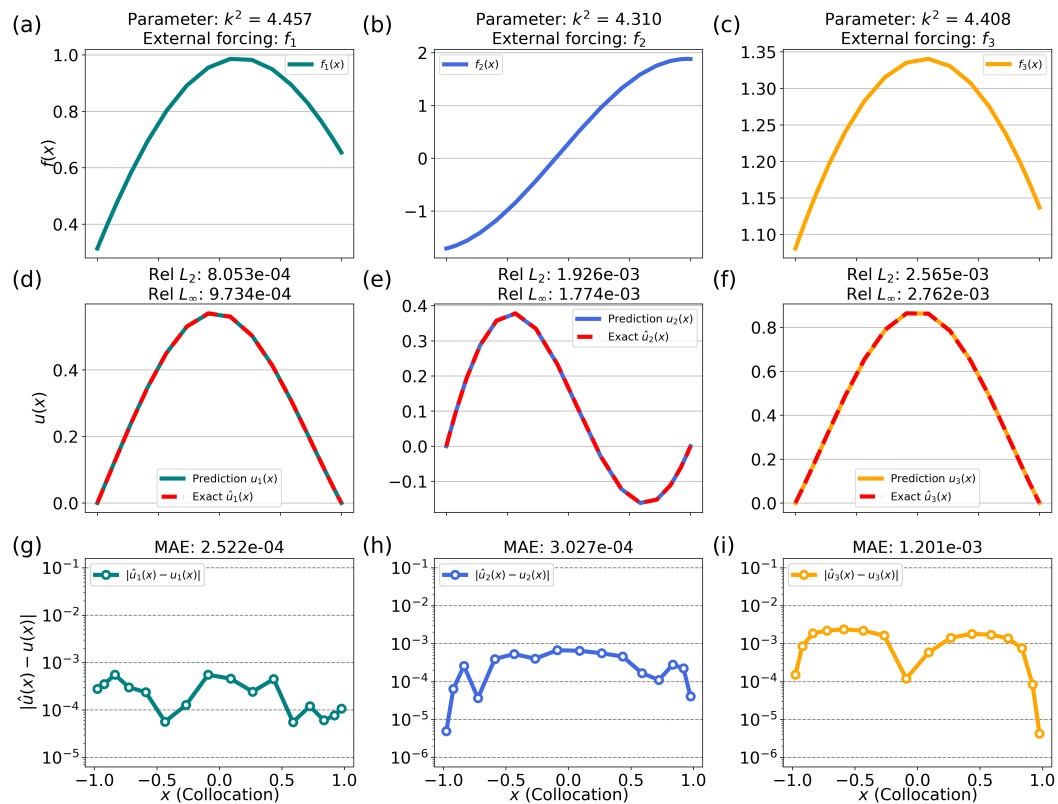

Figure 4: Numerical examples of operator learning with joint parameter and forcing inputs for the Helmholtz equation under Dirichlet boundary conditions. Top: input pairs for the angle network: (a) case 1: a forcing function $f_1$ and a diffusion coefficient $k^2 = 4.457$, (b) case 2: $f_2$ and $k^2 = 4.310$, (c) case 3: $f_3$ and $k^2 = 4.408$. Middle: exact solutions $u_i$ and predicted solutions $\hat{u}_i$ for each case: (d) case 1, (e) case 2, (f) case 3. Bottom: absolute error plots $|\hat{u}_i - u_i|$ on collocation points excluding the boundary points (g) case 1, (h) case 2, (i) case 3.

NVLQS. The exact form of all equations and the full descriptions of their weak formulation are provided in Appendix A.10. Details of the training procedure are also provided in Appendix A.3.

## 4.2 Operator Learning with Joint Parameter and Forcing Inputs

This section introduces operator learning with joint parameter and forcing inputs, a framework in which the learned operator maps both a forcing function and a PDE parameter to the corresponding PDE solution. We demonstrate this approach using the Helmholtz equation with a homogeneous Dirichlet boundary condition:

$$
\begin{aligned}
u_{xx}(x) + k^2 u(x) &= f(x), \quad x \in \Omega := (-1, 1) \\
u(x) &= 0, \quad x \in \partial\,\Omega
\end{aligned}
\tag{12}
$$

To implement operator learning with joint inputs, the input instances are tuples of a forcing function $f^{(i)}$ and a wave number $k^{(i)}$, represented as $\{k^{(i)}, f^{(i)}\}_{i=1}^D$ and mapped to angle parameters $\theta^{(i)} = g(k^{(i)}, f^{(i)}; w)$ via the angle network $g$. Given an input instance $(k, f)$, the spectral matrix for the Helmholtz equation, $A(k)$, is determined by the parameter $k$ and expressed as a linear combination of the fixed stiffness $S$ and mass matrices $M$. We then compute their Pauli decompositions, where some Pauli terms may appear in both

decompositions, i.e., certain $P_{l_1}$ and $P_{l_2}$ represent the same operator.

$$A = S + k^2 M = \sum_{l_1} c_{l_1} P_{l_1} + k^2 \sum_{l_2} c_{l_2} P_{l_2}, \tag{13}$$

Since the fixed matrices $S$ and $M$ are independent of $k$, their Pauli decompositions need only be computed once and can be reused for any value of $k$. Denoting $|\hat{\alpha}\rangle = |\hat{\alpha}(k, f; w)\rangle$, we finally express the numerator of the proposed cost function 7 as:

$$\sum_{l_1} \text{Re}\left(\langle F| P_{l_1} |\hat{\alpha}\rangle\right) + k^2 \sum_{l_2} \text{Re}\left(\langle F| P_{l_2} |\hat{\alpha}\rangle\right) \tag{14}$$

Similarly, the term inside the square root in the denominator of the cost function is given by the following expression, where Pauli terms $P_{l_3}$, $P_{l_4}$, $P_{l_5}$, and $P_{l_6}$ are representing the matrices $S^\dagger S$, $S^\dagger M$, $M^\dagger S$, and $M^\dagger M$, respectively, with their corresponding coefficients:

$$\sum_{l_3} c_{l_3} \langle \hat{\alpha}| P_{l_3} |\hat{\alpha}\rangle + k^2 \sum_{l_4} c_{l_4} \langle \hat{\alpha}| P_{l_4} |\hat{\alpha}\rangle + k^2 \sum_{l_5} c_{l_5} \langle \hat{\alpha}| P_{l_5} |\hat{\alpha}\rangle + k^4 \sum_{l_6} c_{l_6} \langle \hat{\alpha}| P_{l_6} |\hat{\alpha}\rangle \tag{15}$$

Figure 4 presents three representative examples of operator learning with joint parameter and forcing inputs for the Helmholtz equation, using Dirichlet boundary conditions. The top row, shown in Figures 4(a)–(c), presents three input instances, $(k_1, f_1)$, $(k_2, f_2)$, and $(k_3, f_3)$, where the wave numbers $k_i$ are sampled from the uniform distribution $U[4, 5]$ and the forcing functions are linear combinations of trigonometric functions. The middle and bottom rows (Figures 4(d)–(f) and (g)–(i), respectively) present NVQLS predictions versus exact solutions and absolute differences along with MAE values. The predictions exhibit small relative errors (below 0.3%) and MAE values (below $2 \times 10^{-3}$), demonstrating the model's ability to approximate the solution operator across varying parameters. Details on operator learning with joint inputs for the reaction-diffusion equation are provided in Appendix A.10.

### 4.3 Spectral Operator Learning for Two-dimensional PDEs

The remaining sections of the numerical experiments focus on results for various two-dimensional problems. For two-dimensional problems, the solution is expressed as a linear combination of basis functions constructed by the tensor product of the corresponding one-dimensional bases in $x$ and $y$:

$$u(x, y) = \sum_{k,j=0}^{N-2} \alpha_{kj} \phi_k(x) \phi_j(y) \tag{16}$$

Similar to the one-dimensional case described in Section 4.1, NVLQS is trained in an unsupervised manner to predict solutions from forcing function inputs in the two-dimensional setting. The forcing functions for two-dimensional PDEs are also a linear combination of trigonometric functions, and the details of their generation process are described in Appendix 34. As an illustrative example, we consider the two-dimensional reaction–diffusion equation with diffusion coefficient $\epsilon$ under homogeneous Dirichlet boundary conditions, as given in Equation 1. Using the weak formulation, the two-dimensional reaction–diffusion equation is transformed into the following matrix representation.

$$A = -\epsilon \left(S \otimes M + M \otimes S\right) + M \otimes M \in \mathbb{R}^{(N-1)^2 \times (N-1)^2} \tag{17}$$

where $\otimes$ presents the tensor product. The exact representation and corresponding spectral matrices of the Helmholtz and convection–diffusion equation in 2D are fully detailed in Appendix A.10. Figure 5 presents numerical results for two-dimensional PDEs. The top row, shown in Figures 5 (a)–(c), displays the exact solutions for the reaction–diffusion, Helmholtz, and convection–diffusion equations, respectively. The corresponding NVLQS predictions are presented in the middle row, Figures 5 (d)–(f), where relative errors remain below 0.6%. The absolute differences are shown in the bottom row, Figures 5 (g)–(i), with MAE for each case remaining below $2 \times 10^{-4}$.

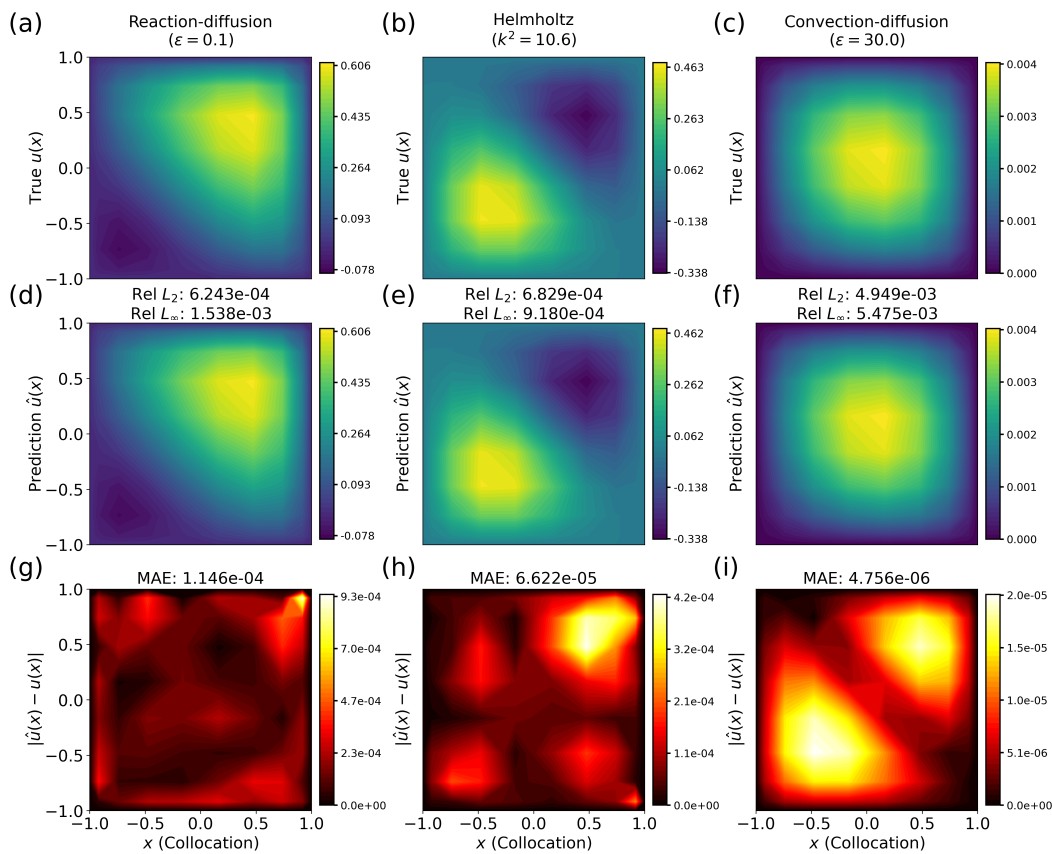

Figure 5: Numerical results for two-dimensional PDEs. Top row: the exact solution $\hat{u}$, middle row: predicted solution $\hat{u}$, bottom row: absolute error $|\hat{u}-u|$, Left column: reaction–diffusion with $\epsilon = 0.1$ (Dirichlet BC), middle column: Helmholtz with $k = 10.6$ (Neumann BC), right column: convection–diffusion with $\epsilon = 30.0$ (Dirichlet BC).

## 5 CONCLUSION

In summary, to address the scalability limitations of classical operator learning in high-resolution and high-dimensional settings, we developed NVQLS which is a quantum-classical hybrid framework based on VQLS. NVQLS defines its cost function in terms of state overlap, which not only reduces the number of required Pauli terms from $O(K^2)$ to $O(K \log K)$ but also enables shallower circuits with fewer quantum gate operations. Since the cost function of NVQLS is motivated by the weak formulation, the training procedure is fully unsupervised. This weak formulation allows NVQLS to incorporate multiple instances of parametric PDEs as model inputs (e.g., forcing functions and PDE parameters) to predict the spectral coefficients of the solution. Leveraging the capabilities of classical neural networks, NVQLS naturally embeds input data into the rotation angles of the quantum circuits, ensuring efficient convergence.

By analyzing the classical and corresponding quantum computational complexity, we suggest a potential quantum advantage in scalability, in terms of the matrix size $K = O(N^d)$ where $N$ is the number of basis and $d$ represents the PDE dimension. Assisted by the use of commutativity-aware grouping in the Hadamard test, the number of required Pauli operators that remains below $O(K \log K)$ in all three cases (reaction–diffusion, Helmholtz, convection–diffusion). The numerical experiments demonstrate that NVQLS accurately predicts PDE solutions from multiple input instances, including PDE parameters and forcing functions. It also generalizes to higher dimensions, where it provides accurate predictions and mathematically enforces boundary conditions.

## Acknowledgments

### Usage of Large Language Models (LLMs)

While an LLM was solely used to assist in sentence editing and code debugging, all outputs were carefully verified and rigorously revised by the author to ensure academic accuracy and appropriateness.

### Reproducibility statement

To ensure the reproducibility of this work, we provide the following details. The detailed derivation of the Legendre–Galerkin weak form used in this work is provided in Appendix A.2. The specific experimental settings and data instance generation scheme are provided in Appendix A.3 and Appendix A.4, respectively. The concrete example for our implementation scheme is provided in Appendix A.5. The figures supporting the complexity analysis of our model are provided in Appendix A.6. In Appendix A.8, we report a simple experiment using a shallow circuit instead of amplitude embedding for hardware-efficient training. In Appendix A.9, we investigate further computational efficiency by considering the perturbed system both analytically and numerically. The code used in the experiments is included in the supplementary material.

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

# A  Appendix

## A.1  Nomenclature

Table 2: Notations and Descriptions

| Notation | Description |
| --- | --- |
| $n$ | Number of qubits |
| $N$ | Number related to the basis function and node point |
| $d$ | Dimension of PDE |
| $L$ | Number of Pauli tensor product matrices for Pauli-decomposition |
| $x_n$ | The $n$-th nodal points on a spatial domain |
| $L_k$ | The Legendre polynomial of degree $k$ |
| $\phi_k$ | The $k$-th basis function |
| $\mathcal{D}$ | Differential operator |
| $\mathcal{B}$ | Boundary operator |
| $u(\cdot)$ | Solution of PDE |
| $\alpha_k$ | The coefficient of $k$-th basis function representing the solution $u(\cdot)$ |
| $\hat{u}(\cdot)$ | Approximated solution |
| $\hat{\alpha}_k$ | The coefficient of $k$-th basis function representing the approximated solution $\hat{u}(\cdot)$ |
| $f(\cdot)$ or $f^{(i)}(\cdot)$ | The ($i$-th) forcing function |
| $\tilde{f}_k$ or $\tilde{f}_k^{(i)}$ | The $k$-th forward transformation of the forcing function $f$ or $f^{(i)}$ |
| $S$ | Stiffness matrix |
| $M$ | Mass matrix |
| $R$ | Convection matrix |
| $F$ or $F^{(i)}$ | The forward-transformed vector (matrix) of forcing vector $f$ or $f^{(i)}$ |
| $D$ | Size of dataset |
| $V(\cdot)$ | Parameterized quantum circuit |
| $\mathcal{L}$ | Cost function |
| $\theta$ | Parameters for quantum circuit (output of angle network) |
| $g(\cdot)$ | Angle network |
| $w$ | Parameters for angle network |
| $\beta$ | Numerator in our loss $\mathcal{L}_{\text{NVQLS}}$ |
| $\gamma$ | Radicand (i.e., the quantity inside the square root) in the denominator in $\mathcal{L}_{\text{NVQLS}}$ |
| $l_V$ | Number of layers of the parameterized quantum circuit $V(\cdot)$ |

## A.2  Derivation of Weak Formulation

In this section, we present the weak formulation in detail. For more details on the spectral methods employed here, we refer the reader to the standard reference (Shen et al., 2011). We begin by examining second-order elliptic partial differential equations on a bounded domain $\Omega \subset \mathbb{R}^n$ under boundary conditions given a PDE parameter $\epsilon > 0$ and an external forcing function $f$:

$$-\epsilon\Delta u + \mathcal{D}(u, \nabla u) = f, \quad x \in \Omega \subset \mathbb{R}^n,$$
$$\mathcal{B}(u, \nabla u) = 0, \qquad x \in \partial\Omega. \tag{18}$$

where $\mathcal{D}$ denotes a possibly nonlinear differential operator and $\mathcal{B}$ represents a boundary operator. For clarity, we first consider the one dimensional reaction–diffusion case where $\mathcal{D}(u) = u$ and $\Omega = (-1, 1)$. For $k = 0, 1, \ldots, N-2$, the Legendre–Galerkin weak formulation is given by

$$\int_\Omega -\epsilon\Delta u(x)\,\phi_k(x)\,dx + \int_\Omega u(x)\,\phi_k(x)\,dx = \int_\Omega f(x)\,\phi_k(x)\,dx. \tag{19}$$

where basis functions are compact combinations of Legendre polynomials $\{L_k\}$:

$$\phi_k = L_k + a_k L_{k+2} + b_k L_{k+2} \tag{20}$$

where the exact boundary conditions, including Dirichlet, Neumann, and mixed types, are mathematically enforced by selecting the coefficients $a_k$ and $b_k$. By representing the solution $u$ as a finite linear combination of the basis functions $\{\phi_k\}_{k=0}^{N-2}$, the weak formulation gives

$$-\epsilon \sum_{j=0}^{N-2} \alpha_j \left( \int_\Omega \phi_j''(x)\phi_j(x)dx \right) + \sum_{k=0}^{N-2} \alpha_j \left( \int_\Omega \phi_j(x)\phi_k(x)dx \right) = \int_\Omega f(x)\,\phi_k(x)\,dx. \quad (21)$$

Then, this produces a linear system $(-\epsilon S + M)\alpha = F$ corresponding to the one-dimensional reaction–diffusion equation where $S$ denote the stiffness matrix, $M$ represent the mass matrix. Their entries are specified as, for $k, j = 0, \ldots, N - 2$:

$$S_{kj} = \int_\Omega \phi_j''(x)\phi_k(x)\,dx, \quad M_{kj} = \int_\Omega \phi_j(x)\phi_k(x)\,dx, \quad (22)$$

with the forward transformed vector given by

$$F = \begin{bmatrix} \tilde{f}_0 & \tilde{f}_1 & \cdots & \tilde{f}_{N-2} \end{bmatrix}^T, \quad \tilde{f}_k = \int_\Omega f(x)\phi_k(x)\,dx, \quad (23)$$

By solving this linear system, one obtains the spectral coefficients

$$\alpha = \begin{bmatrix} \alpha_0 & \alpha_1 & \cdots & \alpha_{N-2} \end{bmatrix}^T. \quad (24)$$

The stiffness matrix $S$ is diagonal and the mass matrix $M$ is symmetric penta-diagonal, whose entries are given by

$$S_{kj} := \int_I \phi_j''\phi_k w\,dx = \begin{cases} (4k+6)b_k, & j = k, \\ 0, & \text{otherwise}, \end{cases} \quad (25)$$

and

$$M_{jk} = M_{kj} := \int_I \phi_j\phi_k w dx = \begin{cases} \frac{2}{2k+1} + a_k^2 \frac{2}{2k+3} + b_k^2 \frac{2}{2k+5}, & j = k, \\ a_k \frac{2}{2k+3} + a_{k+1}b_k \frac{2}{2k+5}, & j = k+1, \\ b_k \frac{2}{2k+5}, & j = k+2, \\ 0, & \text{otherwise}. \end{cases} \quad (26)$$

Similarly, for the one-dimensional convection–diffusion equation with Dirichlet boundary conditions, we define the convection matrix $R$ (given in Equation 27) to represent the first spatial derivative $u_x$:

$$R_{kj} = -R_{jk} = \int_I \phi_j'\phi_k w\,dx = \begin{cases} 2 & k = j+1 \\ -2 & k = j-1 \end{cases} \quad (27)$$

For the two-dimensional reaction–diffusion equation, shown in Equation 1, we employ two-dimensional basis functions constructed as tensor products of one-dimensional basis functions:

$$\{\phi_k(x)\phi_j(y) : k, j = 0, \ldots, N-2\}, \quad (28)$$

where $\phi.(\cdot)$ denotes the one-dimensional basis functions, each formed as a compact combination of Legendre polynomials. Consequently, the predicted solution $\hat{u}(x, y)$ on the two-dimensional domain is expressed as a linear combination of these tensor-product basis functions, with coefficients $\hat{\alpha}_{kj}$:

$$\hat{u}(x, y) = \sum_{k=0}^{N-2} \sum_{j=0}^{N-2} \hat{\alpha}_{kj}\phi_k(x)\phi_j(y). \quad (29)$$

The weak formulation in two dimensions can be expressed as

$$-\epsilon \iint_\Omega \Delta u(x,y)\,\phi_k(x)\phi_j(y)dxdy + \iint_\Omega u(x,y)\,\phi_k(x)\phi_j(y)dxdy = \iint_\Omega f(x,y)\,\phi_k(x)\phi_j(y)dxdy, \quad (30)$$

or equivalently

$$(-\epsilon M \otimes M + S \otimes M + M \otimes S)\hat{\alpha} = F \quad (31)$$

where $\hat{\alpha}$ and $F$ are vectors formed by

$$F = (\tilde{f}_{0,0}, \tilde{f}_{1,0}, \cdots, \tilde{f}_{N-2,0}; \tilde{f}_{0,1}, \cdots, \tilde{f}_{N-2,1}; \tilde{f}_{0,N-2}, \cdots, \tilde{f}_{N-2,N-2})^T \quad (32)$$

and $\otimes$ represents the operation $A \otimes B = (Ab_{ij})_{i,j=0,1,\cdots,N-2}$ (i.e. the tensor product).

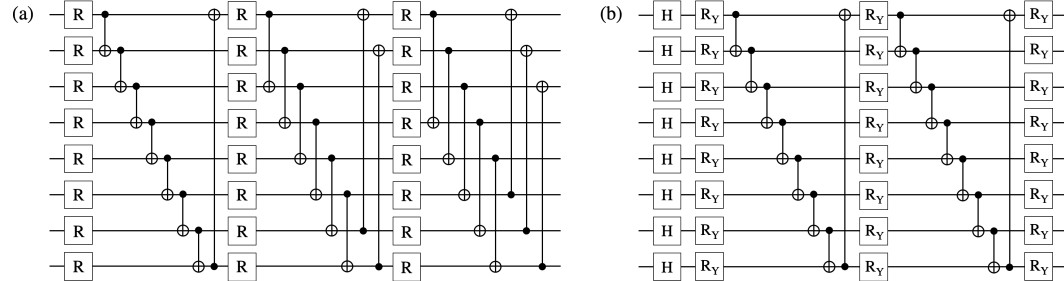

Figure 6: Ansätze used in our study. (a) Strongly Entangling Layer, (b) Hardware efficient RY ansatz.

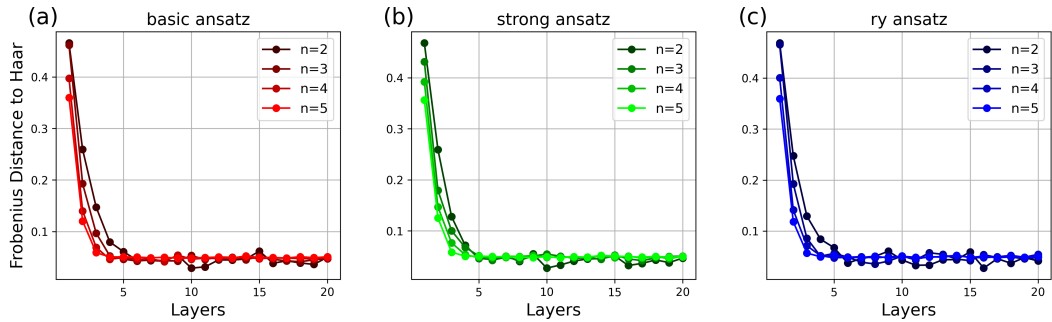

Figure 7: Frobenius distance from Haar distribution. (a) Basic entangling layer, (b) Strongly entangling layer, and (c) Rotation-Y layer

### A.3  Training Details

In this section, we describe the specific structures of each model components discussed in Sec. 3.1, used in our experiments. Our model employed the ansatz $V(\theta)$ to prepare $|\alpha\rangle = V(\theta)|0\rangle$ and utilize the classical neural network $g$ to represent $\theta$ as $\theta = g(w; F)$ with its parameters $w$ for given forcing vector $f$. Furthermore, two types of loss functions are employed in this work.

**Ansatz**   We used two types of ansatz $V(\theta)$: the strongly entanglying circuit and the hardware-efficient RY ansatz. Figure 6 describes each ansatz structure.

The strongly entanglying layer consists of rotation gates $R$ and CNOT gate. The rotation gate $R$ can represent a general rotation by using three parameters $\phi, \theta, \omega$ through two types of single qubit rotation gate $R_Y$ and $R_Z$, defined as $R \equiv R(\phi, \theta, \omega) = R_Y(\phi)R_Z(\theta)R_Y(\omega)$. This ansatz contains strong entanglements where each qubit is entangled with two other qubits with CNOT gates in each layer, employing the different patterns of entanglements per layer. Due to its characteristics, this possess high representation capacity which can be quantified with high expressibility and high entangling capacity. The number of parameters for this ansatz is $(3 \times n \times l_V)$ which is proportional to the number of qubits $n$ and the number of the ansatz layer $l_V$.

The Hardware efficient RY ansatz is composed of Hadamard gates $H$, single qubit rotation gates on y-axis $R_Y$, and CNOT gates. The Hadamard gates are applied first on each qubit, and rotation gates $R_Y \equiv R_Y(\theta)$ are paramterized with $\theta$. In each layer, the nearest neighbour qubits are entangled by CNOT as well as the first and the last qubits. This ansatz is hardware efficient. Moreover, it maps the initial state $|0\rangle$ to real-valued vector $|\alpha\rangle \in \mathbb{R}^{2^n}$ so we don't need to consider the imaginary part when extracting the vector components from $|\alpha\rangle$. The number of parameters for this ansatz is $(n \times l_V)$, also proportional both to $n$ and $l_V$.

**Classical Neural Network** As the angle networks for one-dimensional PDEs, we employ feed-forward (FF) neural networks with Rectified Linear Unit (ReLU) (Agarap, 2019) activation functions. To ensure stable and fast convergence, the L-BFGS optimizer (Liu & Nocedal, 1989) is utilized for the cost function. For the operator learning with joint inputs, a deeper and wider network structure with six FF layers was utilized to enhance the generalization ability of the angle networks. For two-dimensional PDEs, we use a hybrid network combining three convolutional neural network (CNN) layers followed by three FF layers, with the Gaussian error linear units (GELU) (Hendrycks & Gimpel, 2016) activation functions. For the convection–diffusion equation with preconditioner, the adaptive moment estimation (Adam) Kingma & Ba (2017) optimizer is utilized. For hyper-parameter optimization, we adopted a heuristic, progressive approach, starting from a minimal architecture and increasing width or depth only when it improved generalization on validation PDE instances. Due to the shallow quantum circuits in NVQLS, small architectures sufficed for stable training.

Table 3: Details of the structure of the angle network

| PDE | BC | NN | LAYER | OPTIMIZER | ACTIVATION |
|---|---|---|---|---|---|
| RD 1D  16 | Dirichlet | FF | 3 | L-BFGS | ReLU |
| Helmholtz 1D  17 | Neumann | FF | 3 | L-BFGS | ReLU |
| CD 1D 20 | Dirichlet | FF | 6 | L-BFGS | ReLU |
| Helmholtz 1D  4 | Dirichlet | FF | 6 | L-BFGS | ReLU |
| RD 2D 21 | Dirichlet | CNN+NN | 3+3 | L-BFGS | GELU |
| Helmholtz 2D  22 | Dirichlet | CNN+FF | 3+3 | L-BFGS | GELU |
| CD 2D | Dirichlet | CNN+FF | 3+3 | L-BFGS | GELU |
| Helmholtz 1D  18 | Dirichlet | FF | 3 | L-BFGS | ReLU |
| RD 1D 19 | Dirichlet | FF | 3 | Adam ($lr = 10^{-4}$) | ReLU |

Table 4: Details of the structure of the quantum circuit ansatz

| PDE | BC | ANSATZ STRUCTURE | $n$ | DEPTH |
|---|---|---|---|---|
| RD 1D | Dirichlet | StronglyEntanglingLayers | 5 | 12 |
| Helmholtz 1D | Neumann | StronglyEntanglingLayers | 5 | 12 |
| Helmholtz 1D | Dirichlet | StronglyEntanglingLayers | 5 | 12 |
| CD 1D | Dirichlet | StronglyEntanglingLayers | 5 | 12 |
| RD 1D joint | Dirichlet | StronglyEntanglingLayers | 6 | 20 |
| Helmholtz 1D joint | Dirichlet | StronglyEntanglingLayers | 5 | 15 |
| RD 2D | Dirichlet | StronglyEntanglingLayers | 6 | 12 |
| Helmholtz 2D | Dirichlet | StronglyEntanglingLayers | 6 | 12 |
| CD 2D | Dirichlet | StronglyEntanglingLayers | 6 | 20 |

**Loss functions** In the main text, we defined our loss function in Equation 7. For training stability, we also consider an unnormalized cost function $\hat{\mathcal{L}}_{\text{NVQLS}}$ defined as

$$
\hat{\mathcal{L}}_{\text{NVQLS}}(w) = \frac{1}{D} \sum_{i=1}^{D} \left( \sum_{l=1}^{L} \text{Re}\left( \langle F^{(i)} | A_l | \alpha(f^{(i)}; w) \rangle \right) - \sqrt{ \sum_{l,l'=1}^{L} \langle \hat{\alpha}(f^{(i)}; w) | A_{l'}^{\dagger} A_l | \hat{\alpha}(f^{(i)}; w) \rangle } \right)^2 .
$$
(33)

In practice, this loss function often provides better convergence, whereas the fractional form in Equation 7 can sometimes become unstable during training.

**Simulation Details** In experiments, we fixed the number of epochs to 200,000 for both Adam and L-BFGS in order to use exactly the same optimization budget for all experiments

---

**Algorithm 1** Training procedure of NVQLS

---

**Require:** Forcing functions $\{f^{(i)}\}_{i=1}^{D}$ with corresponding forward transformation $\{\tilde{f}^{(i)}\}_{i=1}^{D}$

---

1: Compute the unitary decomposition $A = \sum_{l=1}^{L} c_l A_l$
2: Initialize NVQLS parameters $w$
3: **for** $t = 1$ to $T$ **do**
4:    Evaluate the circuit parameter $\theta^{(i)} = g(f^{(i)}; w)$
5:    Evaluate $\beta_l^{(i)} = \sum_{l=1}^{L} \text{Re} \left( \langle F^{(i)} | A_l | \hat{\alpha}(\theta^{(i)}) \rangle \right)$ where $\hat{\alpha}(\theta^{(i)}) = V(\theta^{(i)})|0\rangle$
6:    Evaluate $\gamma_{l,l'}^{(i)} = \sum_{l=1,l'=1}^{L} \langle \hat{\alpha}(\theta^{(i)}) | A_l^\dagger A_{l'} | \hat{\alpha}(\theta^{(i)}) \rangle$
7:    Evaluate $\mathcal{L}_{\text{NVQLS}}(w)$
8:    Update NVQLS parameter $w$
9: **end for**
10: **return** Trained parameters $w$

---

and keep the comparison between optimizers methodologically fair. We also evaluated the model performance using the recorded metrics at epochs 50k, 100k, 150k, and 200k, and report the corresponding training and test relative errors in Tables A.3 and A.3. Yet, it is worthy to note that, in practical applications, a smaller number of epochs (e.g., 100,000) is sufficient to achieve essentially the same accuracy. Additionally, all simulation trainings are performed using PennyLane (Bergholm et al., 2020), supported by the JAX framework (Bradbury et al., 2018).

Table 5: Training relative $L_2$ error across training epoch

| PDE | BC | EPOCH 50k | EPOCH 100k | EPOCH 150k | EPOCH 200k |
|---|---|---|---|---|---|
| RD 1D  16 | Dirichlet | $\mathbf{1.773 \times 10^{-3}}$ | $1.773 \times 10^{-3}$ | $1.773 \times 10^{-3}$ | $1.773 \times 10^{-3}$ |
| Helm 1D  17 | Neumann | $8.199 \times 10^{-3}$ | $\mathbf{6.277 \times 10^{-3}}$ | $6.277 \times 10^{-3}$ | $6.277 \times 10^{-3}$ |
| CD 1D  19 | Drichlet | $3.954 \times 10^{-3}$ | $3.148 \times 10^{-3}$ | $2.791 \times 10^{-3}$ | $\mathbf{2.598 \times 10^{-3}}$ |
| Helm  18 | Dirichlet | $\mathbf{6.708 \times 10^{-3}}$ | $6.708 \times 10^{-3}$ | $6.708 \times 10^{-3}$ | $6.708 \times 10^{-3}$ |
| RD 2D  21 | Drichlet | $1.021 \times 10^{-3}$ | $7.677 \times 10^{-4}$ | $6.850 \times 10^{-4}$ | $\mathbf{6.490 \times 10^{-4}}$ |
| Helm 2D  22 | Drichlet | $2.481 \times 10^{-3}$ | $\mathbf{1.960 \times 10^{-3}}$ | $1.960 \times 10^{-3}$ | $1.960 \times 10^{-3}$ |
| CD 2D  23 | Drichlet | $5.948 \times 10^{-3}$ | $\mathbf{5.947 \times 10^{-3}}$ | $5.947 \times 10^{-3}$ | $5.947 \times 10^{-3}$ |

Table 6: Test relative $L_2$ error across training epoch

| PDE | BC | EPOCH 50k | EPOCH 100k | EPOCH 150k | EPOCH 200k |
|---|---|---|---|---|---|
| RD 1D  16 | Dirichlet | $\mathbf{5.658 \times 10^{-3}}$ | $5.658 \times 10^{-3}$ | $5.658 \times 10^{-3}$ | $5.658 \times 10^{-3}$ |
| Helm 1D  17 | Neumann | $\mathbf{1.038 \times 10^{-3}}$ | $9.072 \times 10^{-3}$ | $9.072 \times 10^{-3}$ | $9.072 \times 10^{-3}$ |
| CD 1D  19 | Drichlet | $4.624 \times 10^{-3}$ | $3.781 \times 10^{-3}$ | $3.455 \times 10^{-3}$ | $\mathbf{3.279 \times 10^{-3}}$ |
| Helm  18 | Dirichlet | $\mathbf{9.261 \times 10^{-3}}$ | $9.261 \times 10^{-3}$ | $9.261 \times 10^{-3}$ | $9.261 \times 10^{-3}$ |
| RD 2D  21 | Drichlet | $3.666 \times 10^{-3}$ | $3.685 \times 10^{-4}$ | $\mathbf{3.660 \times 10^{-4}}$ | $3.663 \times 10^{-4}$ |
| Helm 2D  22 | Drichlet | $5.596 \times 10^{-3}$ | $\mathbf{5.277 \times 10^{-3}}$ | $5.277 \times 10^{-3}$ | $5.277 \times 10^{-3}$ |
| CD 2D  23 | Drichlet | $8.134 \times 10^{-3}$ | $\mathbf{8.136 \times 10^{-3}}$ | $8.136 \times 10^{-3}$ | $8.136 \times 10^{-3}$ |

A.4   GENERATION OF DATA INSTANCES

Table A.4 details the generation of data instances, including the external forcing function f and PDE parameters (e.g., diffusion parameter $\epsilon$ and wave number $k^2$). For one-dimensional

Table 7: Generation details for the input PDE instances (800 training, 200 testing).

| PDE | $d$ | INSTANCE | $n$ | $N$ | FORCING DISTRIBUTION $(h_1, h_2, m_1, m_2)$ | PARAMETER DISTRIBUTION |
|---|---|---|---|---|---|---|
| RD  16 | 1D | Forcing | 5 | 33 | $U[0,1)$ | 0.1 |
| Helmholtz  17 | 1D | Forcing | 5 | 33 | $U[0,1)$ | 29.4 |
| RD  20 | 1D | Forcing/parameter | 4 | 17 | $U[0,2)$ | $U[0.1, 1.1)$ |
| Helmholtz  4 | 1D | Forcing/parameter | 4 | 17 | $U[0,2)$ | $U[4,5)$ |
| RD  21 | 2D | Forcing | 6 | 9 | $U[0,1)$ | 0.1 |
| Helmholtz  22 | 2D | Forcing | 6 | 9 | $U[0,1)$ | 10.6 |
| CD  19 | 1D | Forcing | 5 | 33 | $U[0,1)$ | 0.05 |

PDEs, the forcing functions are a linear combination of trigonometric functions, according to Equation 10. The coefficients $h_1$, $h_2$, $m_1$ and $m_2$ are drawn from a uniform distribution $U$. Similarly, PDE parameters are also sampled from a uniform distribution. Specifically, for the reaction–diffusion equation 20, diffusion parameters $\{\epsilon^{(i)}\}_{i=1}^{D}$ are drawn from the uniform distribution $U[0.1, 1.1)$, representing a small parameter regime. For the Helmholtz equation 4, wave numbers $\{(k^{(i)})^2\}_{i=1}^{D}$ are sampled from $U[4,5)$. For the two-dimensional PDEs, the forcing functions are constructed as sums of trigonometric functions of $x$ and $y$, with coefficients $h_1^{(i)}, h_2^{(i)}, m_1^{(i)}$, and $m_2^{(i)}$:

$$f^{(i)}(x,y) = h_1^{(i)} \sin(m_1^{(i)}(x+y)) + h_2^{(i)} \cos(m_2^{(i)}(x+y)), \quad i = 1, 2, \cdots, D \qquad (34)$$

### A.5 Commutativity-Based Grouping Method for Loss Evaluation

In this section, we present a concrete example of loss evaluation using the commutativity-based grouping strategy, as discussed in Sectoin 3.1. This strategy constitutes one of the key contributions of our model and leads to a quantum advantage over classical methods.

Consider the operator

$$A = c_1 A_1 + c_2 A_2, \qquad A_1 = Z_1 \otimes Z_2, \qquad A_2 = Z_1 \otimes I_2, \qquad (35)$$

where $\sigma_q$ with $\sigma \in \{X, Y, Z, I\}$ denotes the Pauli operator acting on the $q$-th qubit.

For $\beta^{(i)}$, we need to evaluate $\beta_1^{(i)}$ and $\beta_2^{(i)}$. Since $[A_1, A_2] = 0$, the value of $\beta_2^{(i)}$ can be reconstructed from the measurement outcomes used for $\beta_1^{(i)}$. Specifically, let the probabilities of the computational basis states $|00\rangle, |01\rangle, |10\rangle, |11\rangle$ be $a, b, c, d$, obtained from the measurement associated with $\beta_1^{(i)}$. Then, we obtain

$$\beta_1^{(i)} = a - b - c + d, \qquad \beta_2^{(i)} = a + b - c - d. \qquad (36)$$

For $\gamma^{(i)}$, each component $A_l^\dagger A_{l'}$ can be expressed as a Pauli string. In this example,

$$A_l^\dagger A_{l'} = (|c_1|^2 + |c_2|^2)\, I_1 \otimes I_2 + (c_1 c_2^* + c_1^* c_2)\, I_1 \otimes Z_2. \qquad (37)$$

Thus, the relevant observables are

$$I_1 \otimes I_2, \ I_1 \otimes Z_2. \qquad (38)$$

Since these two operators commute, we can select $I_1 \otimes Z_2$ as the observable to be measured. Consequently, the full quantity can be reconstructed as

$$\gamma^{(i)} = \gamma_{11}^{(i)} + \gamma_{12}^{(i)} + \gamma_{21}^{(i)} + \gamma_{22}^{(i)}, \qquad (39)$$

using only the outcomes of this measurement. The reconstruction follows the same procedure as in the case of $\beta^{(i)}$.

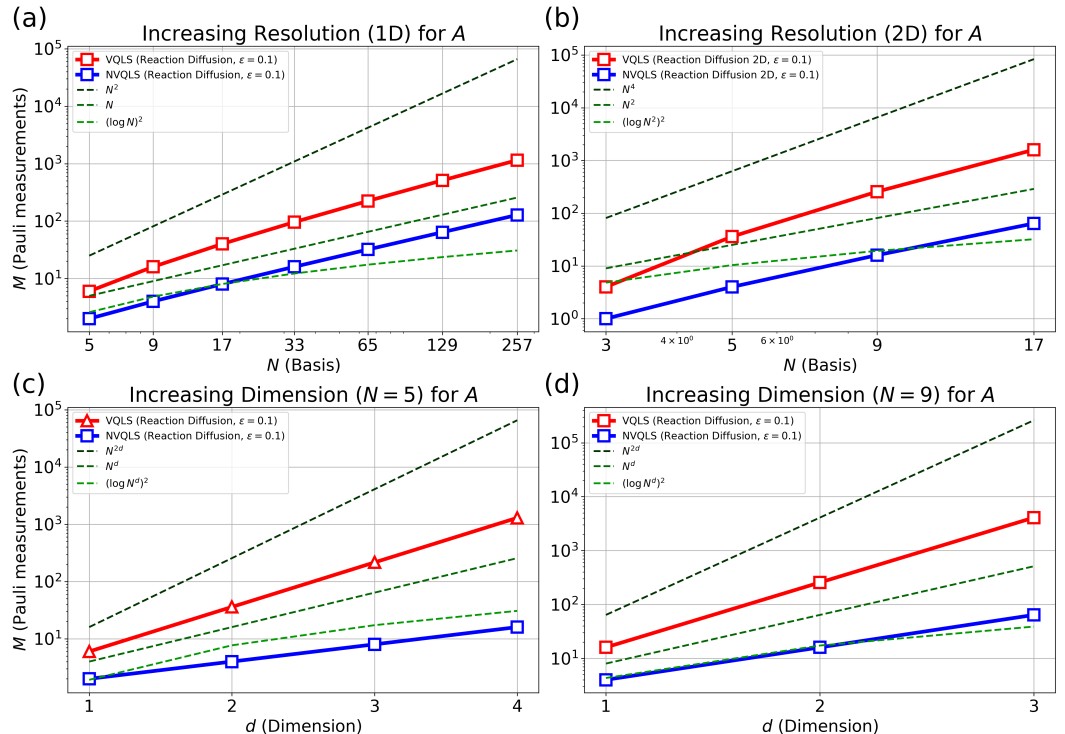

Figure 8: Empirical analysis of the number of required Pauli measurements in NVQLS framework with grouping of commuting measurement operators compared with VQLS for the spectral matrix $A$ of the reaction-diffusion equation ($\epsilon = 0.1$). (a) along the number of basis functions for 1D case, (b) along the number of basis functions for 2D case, (c) along the number of dimension for $N = 5$ case, (d) along the number of dimension for $N = 9$ case.

A.6 Number of Pauli Measurements

This section presents the plots in Figures 10, 11, and 12, which demonstrate the results of the complexity analysis summarized in Tab. 2.

A.7 Performance Metrics

To measure the prediction errors, we employ three metrics: batch-wise mean absolute error (MAE), batch-wise relative $L_2$ error, and batch-wise relative $L_\infty$ error. Given a collection $\{\hat{u}^{(i)}\}_{i=1}^{D}$ of predicted solutions and the corresponding true collection $\{u^{(i)}\}_{i=1}^{D}$, the batch-wise mean absolute error (MAE) is defined by

$$\text{MAE} = \frac{1}{D(N+1)} \sum_{i=1}^{D} \sum_{j=0}^{N} \left| \hat{u}^{(i)}(x_j) - u^{(i)}(x_j) \right|, \tag{40}$$

where $D$ is the number of data instances, $N$ corresponds to the number of basis functions (or spatial points), and $\hat{u}^{(i)}(x_j)$ and $u^{(i)}(x_j)$ are the predicted and true solutions at the collocation point $x_j$, respectively. The batch-wise relative $L_2$ error is given by

$$\text{RelL}_2 = \frac{1}{D} \sum_{i=1}^{D} \frac{\|\hat{u}^{(i)} - u^{(i)}\|_2}{\|u^{(i)}\|_2} = \frac{1}{D} \sum_{i=1}^{D} \frac{\sqrt{\int |\hat{u}^{(i)} - u^{(i)}|^2 dx}}{\sqrt{\int |u^{(i)}|^2 dx}}, \tag{41}$$

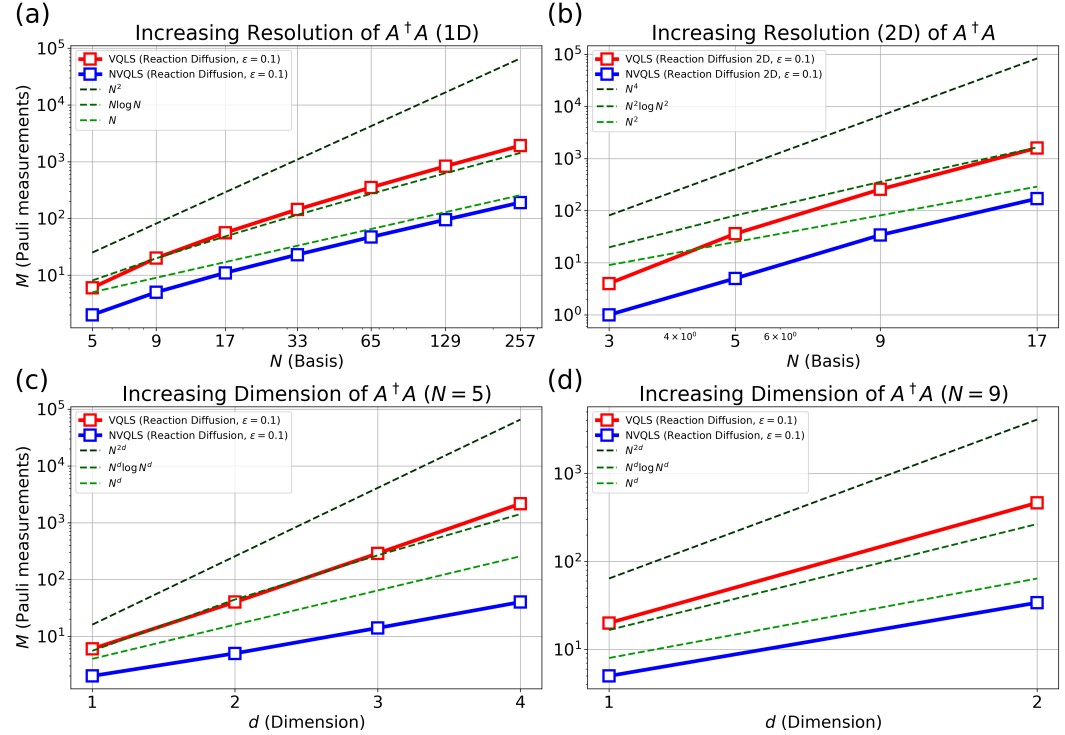

Figure 9: Empirical analysis of the number of required Pauli measurements in NVQLS framework with grouping of commuting measurement operators compared with VQLS for the spectral matrix $A^\dagger A$ of the reaction-diffusion equation ($\epsilon = 0.1$). (a) along the number of basis functions for 1D case, (b) along the number of basis functions for 2D case, (c) along the number of dimension for $N = 5$ case, (d) along the number of dimension for $N = 9$ case.

where the integrals are computed using the Legendre–Gauss–Lobatto (LGL) quadrature. Finally, the batch-wise relative $L_2$ error is defined by

$$\text{RelL}_\infty = \frac{1}{D} \sum_{i=1}^{D} \frac{\|\hat{u}^{(i)} - u^{(i)}\|_\infty}{\|u^{(i)}\|_\infty} = \frac{1}{D} \sum_{i=1}^{D} \sum_{j=0}^{N} \frac{\max_j |\hat{u}^{(i)}(x_j) - u^{(i)}(x_j)|}{\max_j |u^{(i)}(x_j)|}. \tag{42}$$

## A.8 Hardware-Efficient Training

We present a hardware-efficient, circuit-parameterized training scheme to validate our framework on near-term quantum devices. While the main text employs amplitude embedding, its practical implementation on current hardware is constrained by circuit depth. To address this limitation, we adopt a shallow state-preparation routine that uses only single-qubit $R_y$ rotations to construct the forward-transformed forcing vector $F$. This approach substantially reduces circuit depth, making it more suitable for noisy intermediate-scale quantum (NISQ) hardware.

We define an angle vector $\boldsymbol{\theta} = (\theta_1, \ldots, \theta_n)$ and prepare the product state

$$|\psi(\boldsymbol{\theta})\rangle = \bigotimes_{i=1}^{n} R_y(\theta_i)|0\rangle \quad \text{with} \quad R_y(\theta) = \exp\!\left(-\tfrac{i}{2}\theta Y\right), \tag{43}$$

where $Y$ denotes the Pauli-$Y$ operator. Let $\boldsymbol{a}(\boldsymbol{\theta}) \in \mathbb{R}^{2^n}$ denote the real amplitudes of $|\psi(\boldsymbol{\theta})\rangle$ in the computational basis. We set

$$F_k = a_k(\boldsymbol{\theta}), \qquad k = 1, \ldots, N-1, \tag{44}$$

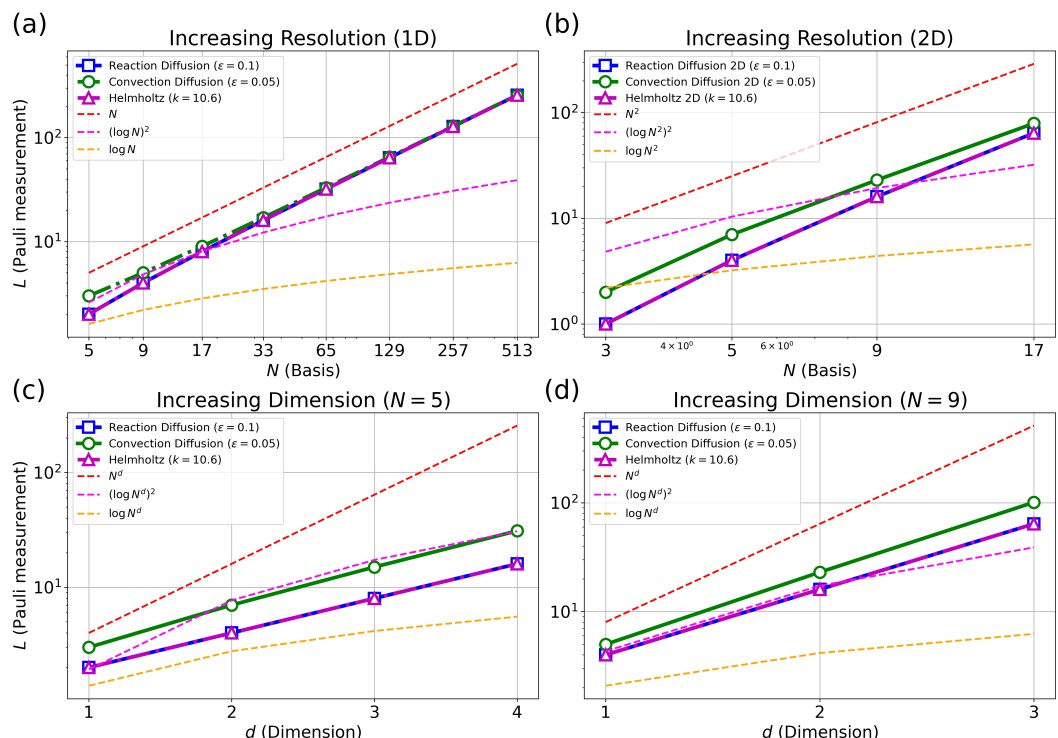

Figure 10: Empirical analysis of the number of Pauli measurements using commutativity-aware grouping. Top: Number of Pauli measurements for increasing resolution (as a function of the number of basis functions $N$): (a) one-dimensional case, (b) Two-dimensional case. Bottom: number of Pauli measurements for increasing dimension of PDEs (as a function of the dimension $d$ of PDEs): (c) $N = 5$, (d) $N = 9$.

so that $F \in \mathbb{R}^{N-1}$. The dataset is generated classically by sampling $\boldsymbol{\theta}$, computing $F$, and storing the resulting pairs $(\boldsymbol{\theta}, F)$. When executing on quantum hardware, $F$ is not loaded via amplitude embedding; instead, the shallow circuit specified by the stored $\boldsymbol{\theta}$ is executed.

We evaluate on the 1D Helmholtz equation with Dirichlet boundary conditions using a 4-qubit system representing 18 nodes and a discrete angle set $\theta_i \in \{0, \frac{\pi}{2}, \pi, \frac{3\pi}{2}\}$. We compare NVQLS to a classical ULGNet on 10,000 samples (train=8,000, test=2,000). Our framework employs a fully connected neural network with 11,080 parameters, while the ULGNet baseline uses a similar network with 12,688 parameters. Figure 13 (a) shows the average test $L_2$ and $L_\infty$ errors, where our framework attains lower test errors. Figures 13 (b)-(d) visualize test solutions from our framework corresponding to the best, median, and worst relative error cases. The visualizations suggest that our framework generalizes effectively to previously unseen instances.

## A.9 TRUNCATION METHOD

For further computational efficiency, we approximate the target system $A\alpha = F$ by truncating the Pauli-decomposed terms of $A$, yielding $\tilde{A}\tilde{\alpha} = F$. In this approach, we retain only $L'$ Pauli terms, $\tilde{A} = \sum_{l=1}^{L'} c_l A_l$, instead of the full $L$ terms, $A = \sum_{l=1}^{L} c_l A_l$, with $L' < L$. This truncation reduces the computational cost of the loss evaluation.

We set the truncation level such that the relative $l_2$ error between the reference solution $u$ (with the full operator $A$) and the approximated solution $\tilde{u}$ (with $\tilde{A}$) is on the order of $10^{-3}$. To this end, we rank the Pauli components by coefficient magnitude $|c_l|$ and retain the largest terms until the error criterion $\|u - \tilde{u}\|_2 / \|u\|_2 \sim O(10^{-3})$ is met.

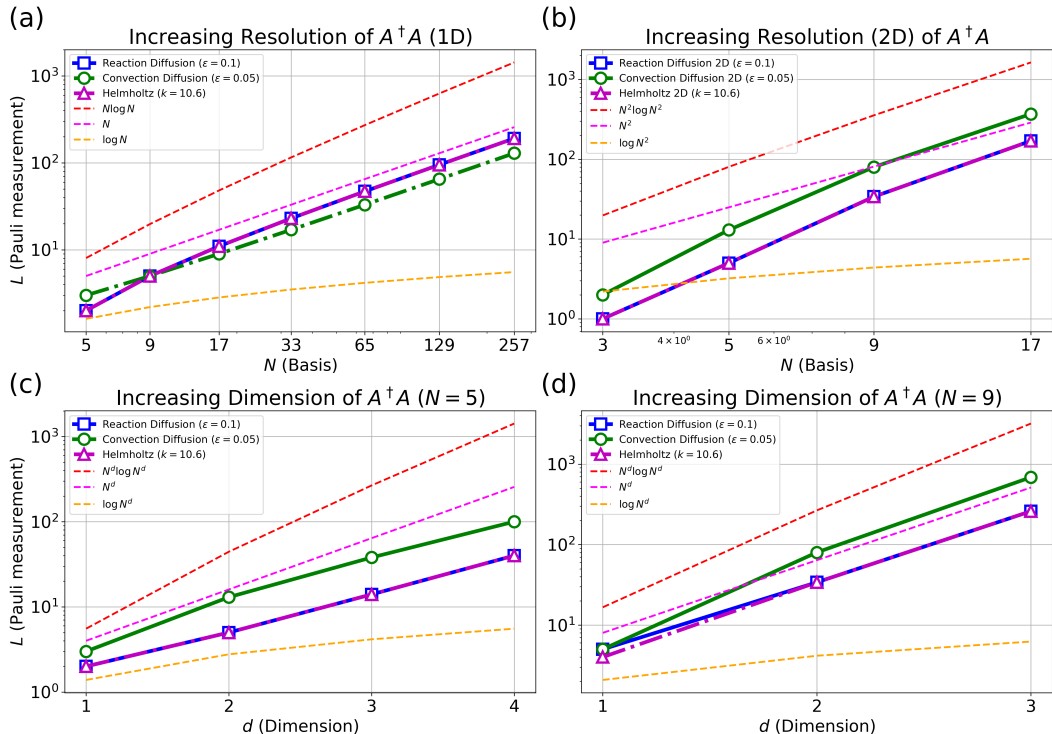

Figure 11: Empirical analysis of the number of Pauli measurements of $A^\dagger A$ using commutativity-aware grouping. Top: number of Pauli measurements of $A^\dagger A$ for increasing resolution (as a function of the number of basis functions $N$): (a) one-dimensional case, (b) two-dimensional case. Bottom: Number of Pauli measurements of $A^\dagger A$ for increasing dimension of PDEs (as a function of the dimension $d$ of PDEs): (c) $N = 5$, (d) $N = 9$.

We applied this truncation method to the 1D Helmholtz case with Neumann boundary conditions and observed a reduction in the computational cost of loss evaluation. The simulation was conducted under the same experimental settings as the non-truncated case in Figure 18.

Figure 14 and Figure 15 summarize the results. In Figure 14, $(a)$ shows the scaling behavior of the number of Pauli terms in the truncated operator $\tilde{A}$ compared to the original $A$, with the complexity reduced from $O(N \log N)$ to $O(N)$. Figure 14 $(b)$ and $(c)$ presents the corresponding number of Pauli measurements required for our loss evaluation in the calculation of $\beta$ and $\gamma$, comparing the truncated $\tilde{A}$ with $A$, respectively. Here, the scaling is reduced from $O(N)$ to $O(\log N)$ for both $\beta$ and $\gamma$, leading to a combined reduction from $O(N)$ to $O(\log N)$.

Figure 15 present simulation results for the truncated 5-qubit system. While the performance is somewhat worse than in the full case, this indicates that a truncation threshold of $10^{-3}$ is insufficient to obtain a solution close to the exact one. Nevertheless, there remains room to identify a more suitable threshold that balances acceptable accuracy with reduced loss evaluation cost, thereby yielding further quantum advantages.

### A.10 Detailed Numerical Experiments

In this section, we provide the exact form of the PDEs, their corresponding spectral matrices, and the numerical results of our experiments.

**Reaction–diffusion Equation with Dirichlet Boundary** We start with the one-dimensional reaction–diffusion equations (RDEs) with a diffusion coefficient $\epsilon$ under the

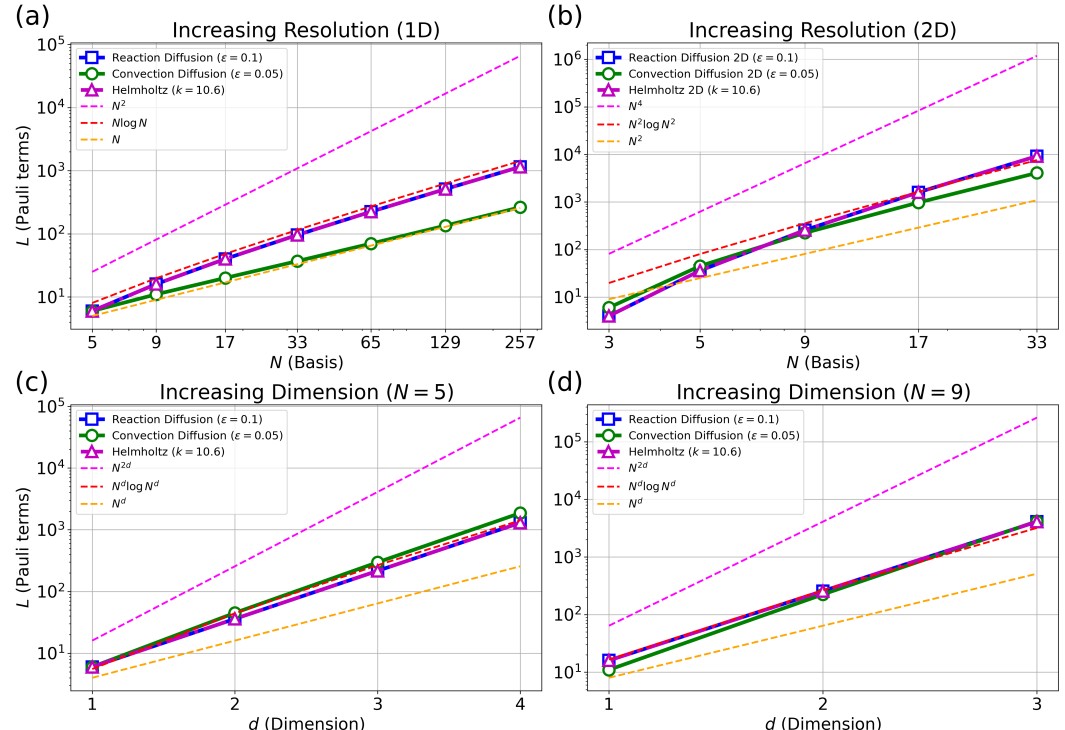

Figure 12: Empirical analysis of the number of Pauli terms of $A$ without commutativity-aware grouping. Top: number of Pauli terms of $A$ for increasing resolution (as a function of the number of basis functions $N$): (a) one-dimensional case, (b) two-dimensional case. Bottom: number of Pauli terms of $A$ for increasing dimension of PDEs (as a function of the dimension $d$ of PDEs): (c) $N = 5$, (d) $N = 9$.

homogeneous Dirichlet boundary condition 45:

$$\begin{aligned} -\epsilon u_{xx}(x) + u(x) &= f(x), \quad x \in \Omega := (-1, 1) \\ u(x) &= 0, \quad x \in \partial\,\Omega \end{aligned} \tag{45}$$

As derived in Appendix A.2, the corresponding matrix is given by

$$A = -\epsilon S + M. \tag{46}$$

Figure 16 shows numerical results on a reaction–diffusion equation with a diffusion coefficient $\epsilon = 0.1$. Figure 16 (a)–(b) shows a representative example of the NVLQS prediction $\hat{u}(x)$ to the true solution $u(x)$ with the absolute error $|\hat{u}(x) - u(x)|$ over the spatial domain $\Omega = (-1, 1)$. This figure shows that the predicted solution $\hat{u}(x)$ is sufficiently close to the exact solution $u(x)$. The relative $L_2$ error $(4.974 \times 10^{-3})$ and relative $L_\infty$ error $(5.611 \times 10^{-3})$ both remain below 0.6%, with a MAE of $1.396 \times 10^{-3}$. Figure 16,(c) shows that the training and test losses decrease smoothly, indicating stable convergence. As illustrated in Figure 16 (d), the batch mean relative $L_2$ and $L_\infty$ errors steadily decrease as the loss decreases, suggesting that the chosen loss function effectively aligns with the intended error objectives.

**Helmholtz Equation with Neumann Boundary** To demonstrate the ability of the proposed method to handle various boundary conditions, we conduct a numerical experiment on the Helmholtz equation with homogeneous Neumann boundary conditions for representing a wave propagation problem with a wave number $k^2$:

$$\begin{aligned} u_{xx}(x) + k^2 u(x) &= f(x), \quad x \in \Omega := (-1, 1) \\ u_x(x) &= 0, \quad x \in \partial\,\Omega. \end{aligned} \tag{47}$$

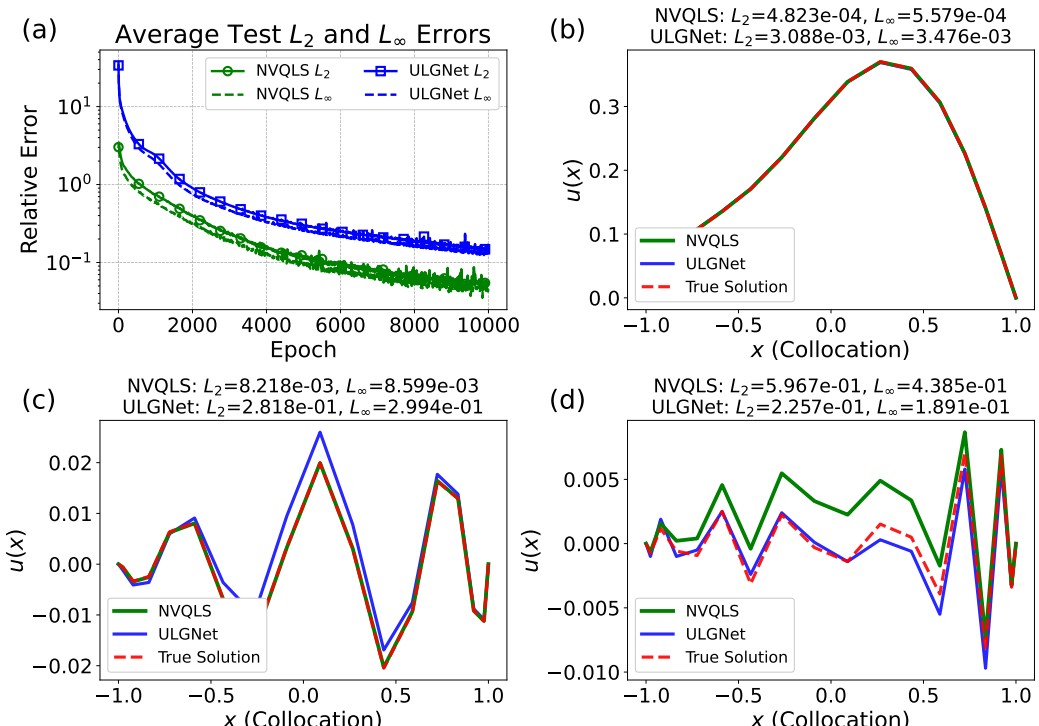

Figure 13: Comparison with a classical ULGNet on the 1D Helmholtz equation with Dirichlet boundary conditions. (a) average test $L_2$ and $L_\infty$ errors over the 2,000 test samples; (b)–(d) representative test solutions from NVQLS at the best/median/worst relative error cases, shown alongside ULGNet and the ground truth. NVQLS uses a fully connected network with 11,080 parameters (4-qubit setting representing 18 nodes), while the ULGNet baseline uses 12,688 parameters.

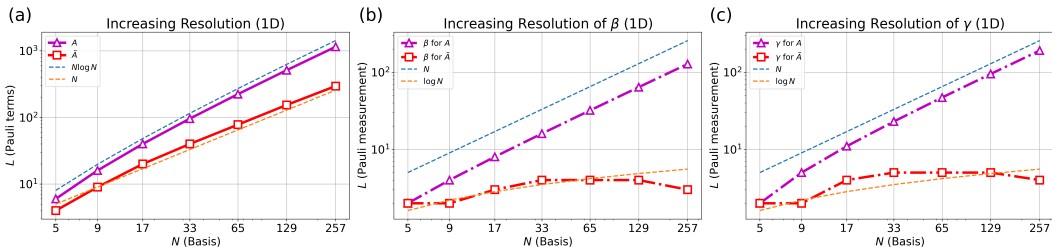

Figure 14: Empirical comparison of $A$ and the truncated $\tilde{A}$ with respect to the number of Pauli terms and Pauli measurements required for loss evaluation ($\beta$ and $\gamma$) in the 1D Helmholtz problem with the Neumann boundary conditions.

The corresponding matrix of spectral methods is $A = S + k^2 M$, a linear combination of the stiffness matrix $S$ and the mass matrix $M$. Here, the homogeneous Neumann boundary condition is strongly enforced by constructing basis functions as

$$\phi_k(x) = L_k(x) - \frac{k(k+1)}{(k+2)(k+3)} L_{k+2}(x). \tag{48}$$

Figure 17 illustrates numerical results of a numerical experiment on the Helmholtz equation with wave number $k^2 = 15.2$. Figure 17 (a) presents the exact solution alongside the NVLQS prediction. The relative error remains below 0.4%, indicating a close agreement between

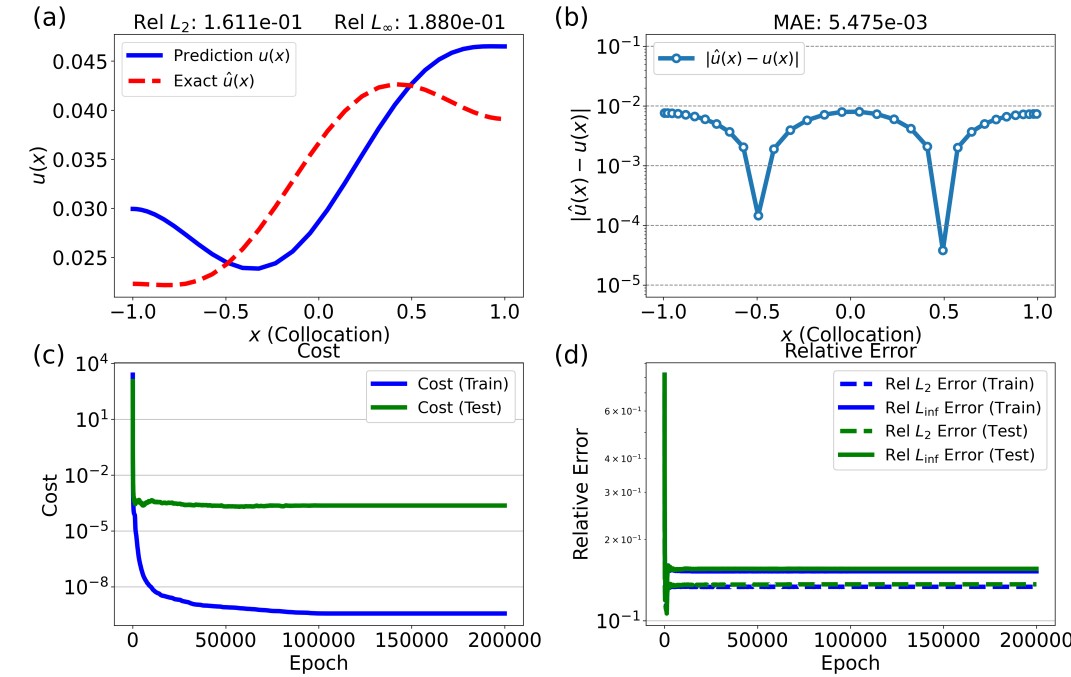

Figure 15: Simulation results for the truncated 5-qubit system with truncation threshold $10^{-3}$ in the 1D Helmholtz problem with the Neumann boundary condition.

the two. Figure 17 (b) displays the absolute error, with a mean absolute error (MAE) of $1.572 \times 10^{-4}$. Figure 17 (c)–(d) illustrates the training and test loss curves, together with their batch-mean relative errors. As the training cost decreases, the relative errors also decline. Although the test loss curve begins to increase after 50,000 epochs, the test relative error continues to decrease, justifying the continuation of training.

**Helmholtz Equation with Dirichlet Boundary** Now, we focus on the homogeneous Helmholtz equation with another boundary condition–Dirichlet BC. The exact form of the equations is given in Equation 12. The corresponding matrix is $A = S + k^2 M$, which is the same matrix of Neumann BC. To strongly enforce Dirichlet boundary condition, the basis functions are constructed as illustrated in Equation 11.

Given a wave number $k^2 = 29.4$ Figure 18 (a) compares the exact solution with the model's prediction, showing a relative error below 0.3%, which confirms their strong consistency. The corresponding absolute error is depicted in Figure 18 (b), yielding a mean absolute error (MAE) of $5.827 \times 10^{-5}$. Figures 18 (c)–(d) track the training and test dynamics, including both the loss curves and the relative errors. A steady decrease in relative error accompanies the reduction of training cost. Even though the test loss begins to rise early, the continued decrease in the test relative error indicates that further training remains beneficial for achieving better solution accuracy.

**Convecion-diffusion Equation with Dirichlet Boundary** We now focus on the convection–diffusion equation, characterized by a convection velocity $v$ and a small diffusion coefficient $\epsilon$:

$$-\epsilon u_{xx}(x) + v u_x(x) = f(x), \quad x \in \Omega := (-1, 1),$$
$$u(x) = 0, \quad x \in \partial \Omega. \tag{49}$$

The linear system $(-\epsilon S + vR)\alpha = F$, constructed by the weak formulation, involves the convection matrix $R$ to represent the first spatial derivative $u_x$ shon in Equation 27. The matrix $R$ is asymmetric and can be singular, which may induce numerical instability during optimization. As a remedy, incomplete LU (ILU) preconditioning is applied to transform the spectral method matrix into a better-conditioned form, ensuring stable convergence.

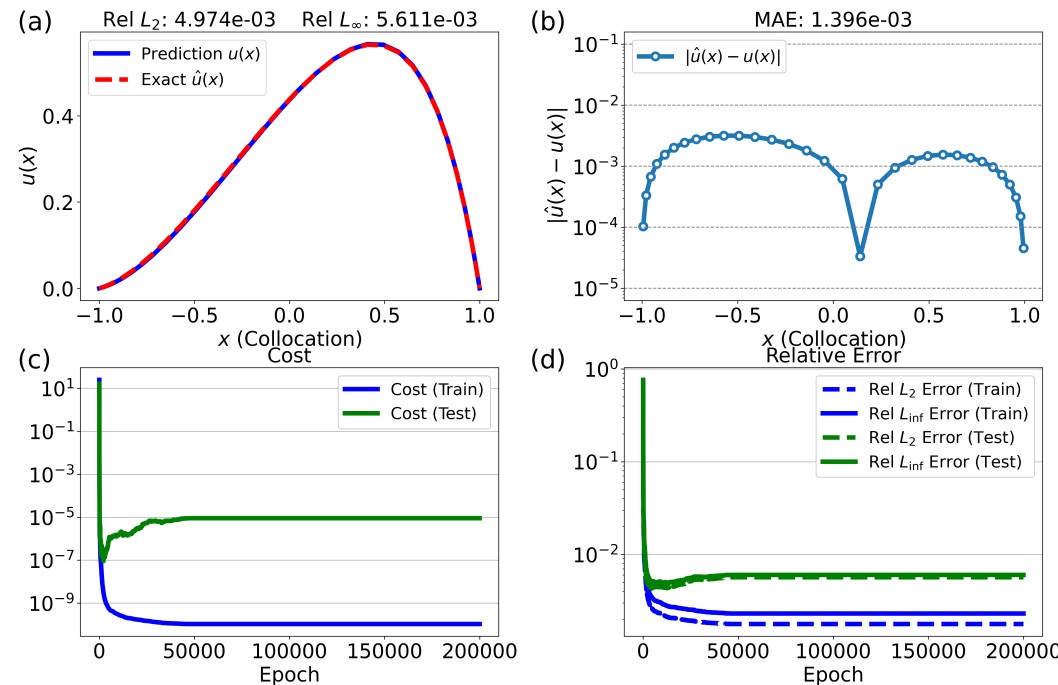

Figure 16: Numerical results for the one-dimensional reaction–diffusion equation with homogeneous Dirichlet boundary conditions and diffusion coefficient $\epsilon = 0.1$; Top row: (a) predicted solution $\hat{u}$ versus exact solution $u$, (b) absolute error $|\hat{u} - u|$. Bottom row: (c) batch-wise training and test losses, (d) batch-wise relative $L_2$ and $L_\infty$ errors over epochs.

With a small diffusion coefficient $\epsilon = 0.05$, Figure 19 (a) confirms that the model prediction aligns closely with the exact solution, maintaining a relative error below 0.4%. This accuracy is attributed to the use of an ILU preconditioner, which, when applied to both sides, improves the conditioning of the spectral system matrix. The corresponding absolute error is reported in Figure 19 (b), with a MAE value of $1.145 \times 10^{-3}$. Figures 19 (c)–(d) further depict the evolution of training and testing, including both loss curves and relative error metrics.

**Operator Learning with Joint Inputs for Reaction-diffusion Equation** To evaluate the capability of our model in learning PDE operators with joint inputs of parameters and forcing terms, we present numerical experiments on the reaction–diffusion equation, whose exact form is provided in Equation 45. By leveraging classical neural networks, each pair of diffusion coefficients and forcing terms is mapped to the circuit parameters through the angle network:

$$\theta^{(i)} = g(\epsilon^{(i)}, f^{(i)}; w). \tag{50}$$

Figure 20 presents the results of operator learning combining parameter and forcing inputs for the reaction–diffusion equation under Dirichlet boundary conditions. The figure is structured into three rows: the top row (a)–(c) displays three distinct input configurations $(\epsilon_1, f_1)$, $(\epsilon_2, f_2)$, and $(\epsilon_3, f_3)$, where the diffusion coefficients $\epsilon_i$ are sampled from $U[0.1, 1.1]$ and forcing functions utilize a linear combination of trigonometric terms. The middle and bottom rows (d)–(f) and (g)–(i) compare the NVQLS predictions with their exact counterparts and illustrate the resulting absolute differences alongside MAE values. The model consistently achieves highly accurate approximations across varying parameter regimes, substantiated by relative errors below 0.07% and MAE values under $2 \times 10^{-4}$.

**Two-dimensional Reaction-diffusion Equations with Dirichlet Boundary** We now extend our study to two-dimensional reaction–diffusion equations with a diffusion coefficient

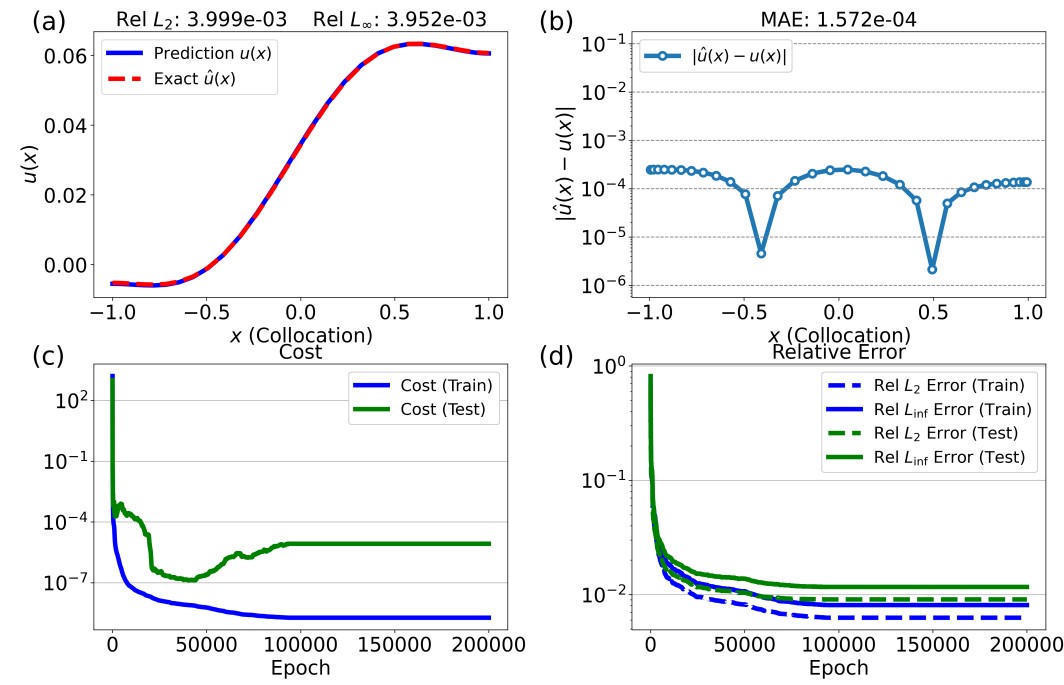

Figure 17: Numerical results for the one-dimensional Helmholtz equation with the homogeneous Neumann boundary condition and a wave number $k^2 = 15.2$; Top: (a) example of the predicted solution $\hat{u}$ compared to the exact solution $u$, (b) absolute error between $\hat{u}$ and $u$. Bottom: (c) batch-wise training and test losses, (d) batch-wise relative $L_2$ and $L_\infty$ errors over epochs.

$\epsilon$, subject to homogeneous Dirichlet boundary conditions:

$$-\epsilon \Delta u(x, y) + u(x, y) = f(x, y), \quad (x, y) \in \Omega$$
$$u(x, y) = 0, \quad (x, y) \in \partial \Omega \tag{51}$$

Using the weak formulation, the two-dimensional reaction––diffusion equation is transformed into the following matrix representation shown in Equation 17. Note that the size of the spectral matrix $A$ grows exponentially in the two-dimensional case compared to the one-dimensional case.

With a diffusion parameter $\epsilon = 0.1$, Figure 21 (a)–(c) presents a representative NVQLS prediction, the corresponding exact solution, and the pointwise absolute error $|\hat{u}(x, y) - u(x, y)|$, respectively. The relative errors remain below 0.2%, indicating that NVQLS closely reproduces the numerical solution. The training and test loss curves are displayed in Figure 21 (d). In this experiment, although the test loss curve rises at an early stage, the relative errors continue to decrease, justifying the continuation of training.

**Two-dimensional Helmholtz Equations with Dirichlet Boundary** We now present numerical results for the two-dimensional Helmholtz equation. The specific form of the equation is given below, characterized by the wave number $k^2$:

$$\Delta u(x, y) + k^2 u(x, y) = f(x, y), \quad (x, y) \in \Omega$$
$$u(x, y) = 0, \quad (x, y) \in \Omega \tag{52}$$

The two-dimensional Helmholtz equation corresponds to a linear system involving matrices expressed as tensor products of one-dimensional matrices:

$$\left(S \otimes M + M \otimes S + k^2 \left(M \otimes M\right)\right) \alpha = F \tag{53}$$

Figure 22 presents the exact solution, the NVQLS prediction, and their absolute difference. Figures 22(a)–(c) demonstrate a close match, with relative errors remaining below 0.1%.

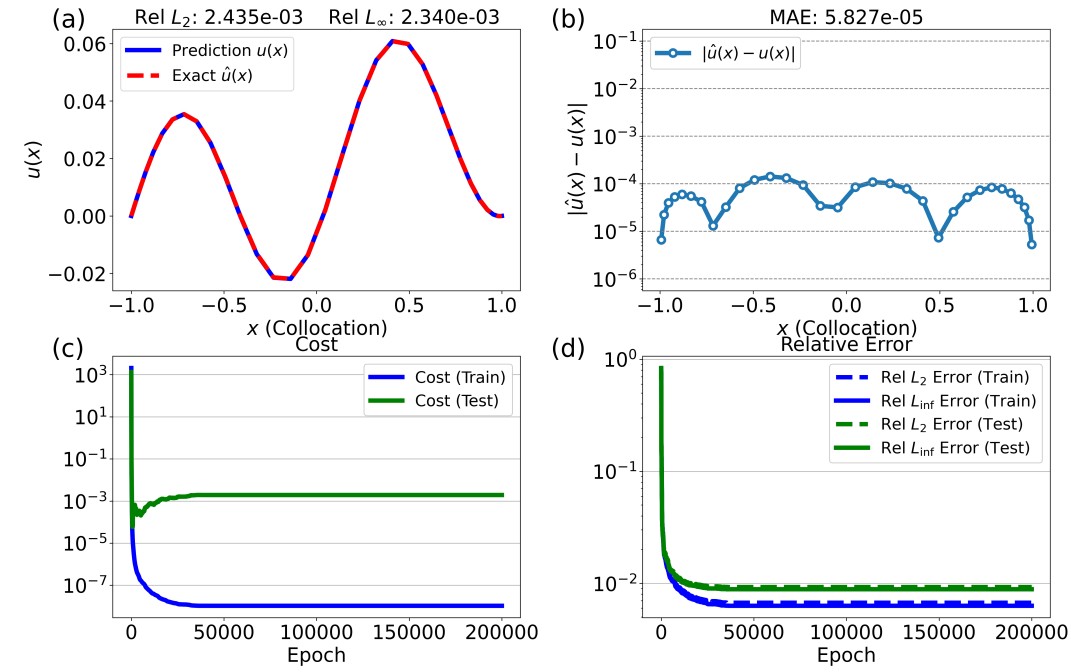

Figure 18: Numerical results for the one-dimensional Helmholtz equation with the homogeneous Dirichlet boundary condition and a wave number $k^2 = 29.4$. Top: (a) example of the predicted solution $\hat{u}$ compared to the exact solution $u$, (b) absolute error between $\hat{u}$ and $u$. Bottom: (c) batch-wise training and test losses, (d) batch-wise relative $L_2$ and $L_\infty$ errors over epochs.

The convergence is further supported by subfigures (d) and (e), where (d) presents the training and test loss curves, and (e) shows the steady decrease in batch-mean relative $L_2$ and $L_\infty$ errors throughout training.

**Two-dimensional Convection-diffusion Equations with Dirichlet Boundary**  We finally explain the convection–diffusion equations with Dirichlet BC in the two dimension, whose exact form is given by

$$-\epsilon \Delta u(x,y) + \nu \cdot \nabla u(x,y) = f(x,y), \quad (x,y) \in \Omega$$
$$u(x,y) = 0, \quad (x,y) \in \Omega \tag{54}$$

From the weak formulation, the corresponding matrix can be written as

$$A = -\epsilon \left( S \otimes M + M \otimes S \right) + \nu_1 \, R \otimes M + \nu_2 \, M \otimes R^T \tag{55}$$

Figure 23 compares the exact solution, the NVQLS prediction, and the pointwise absolute error for the convection–diffusion equation with a diffusion parameter $\epsilon = 30.0$. Panels (a)–(c) reveal a strong agreement between the prediction and the exact solution, with relative errors around 1% and a mean absolute error (MAE) of $1.241 \times 10^{-5}$. Convergence behavior is further illustrated in panels (d) and (e). The evolution of training and test losses is depicted in (d) , while (e) shows that the batch-mean relative $L_2$ and $L_\infty$ errors steadily decrease over the course of training.

**Operator Learning with Joint Inputs for Two-dimensional Helmholtz Equation**

**Operator Learning with Joint Inputs for Two-dimensional Reaction-diffusion Equation**

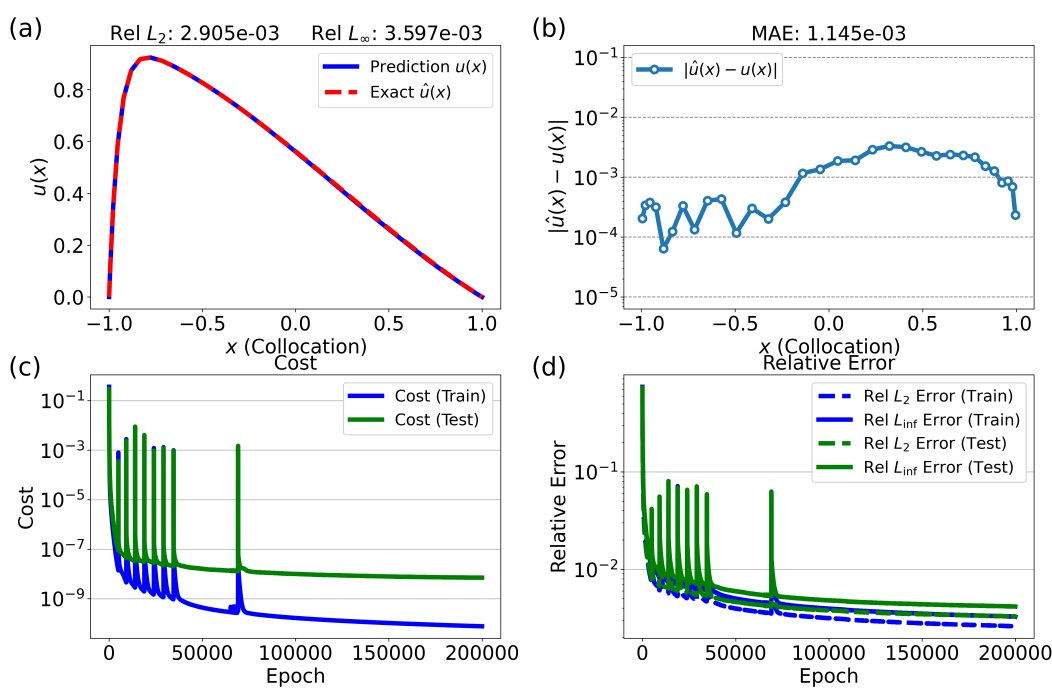

Figure 19: Numerical results for the one-dimensional convection–diffusion equation with the homogeneous Dirichlet boundary condition and with a diffusion coefficient $\epsilon = 0.05$. Top: (a) Example of the predicted solution $\hat{u}$ compared to the exact solution $u$, (b) Absolute error between $\hat{u}$ and $u$. Bottom: (c) Batch-wise training and test losses, (d) Batch-wise relative $L_2$ and $L_\infty$ errors over epochs.

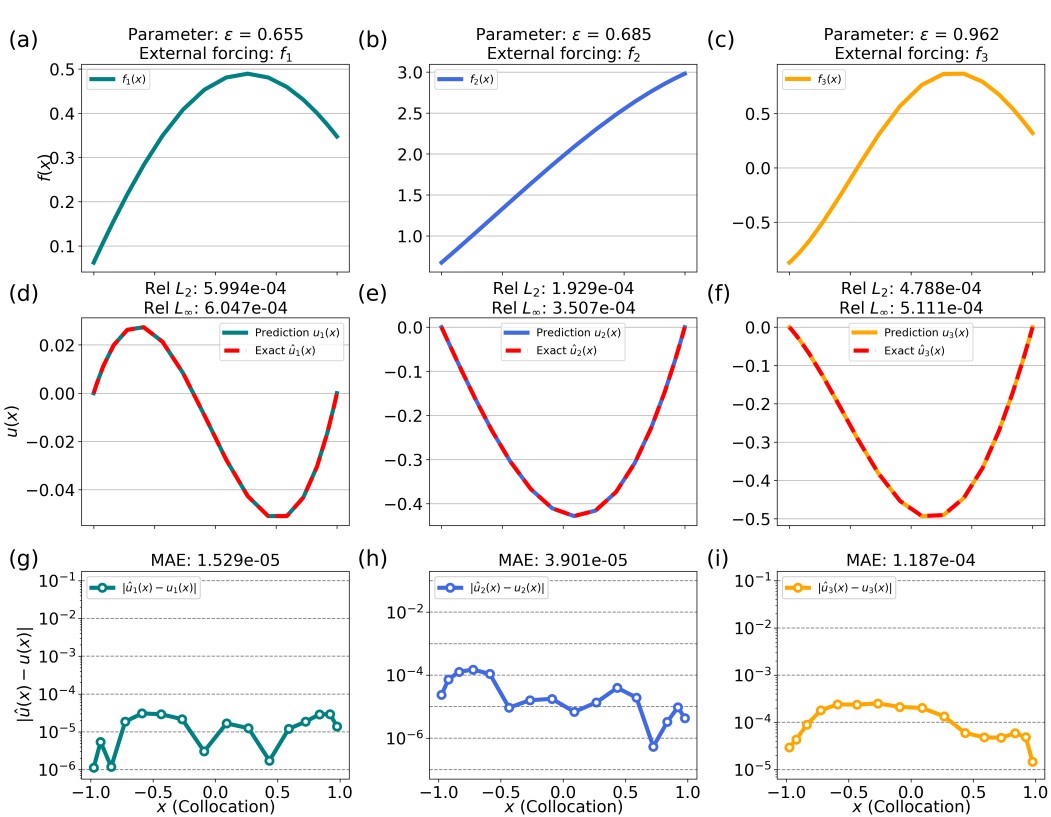

Figure 20: Numerical results for the reaction–diffusion equation with homogeneous Dirichlet boundary conditions. Top: input pairs for the angle network. (a) case 1: a forcing function $f_1$ and a diffusion coefficient $\epsilon = 0.360$, (b) case 2: $f_2$ and $\epsilon = 0.205$, (c) case 3: $f_3$ and $\epsilon = 0.458$. Middle: exact solutions $u^{(i)}$ and predicted solutions $\hat{u}_i$ for each case: (d) case 1, (e) case 2, (f) case 3. Bottom: absolute error plots $|\hat{u}_i - u_i|$ on collocation points excluding the boundary points: (g) case 1, (h) case 2, (i) case 3.

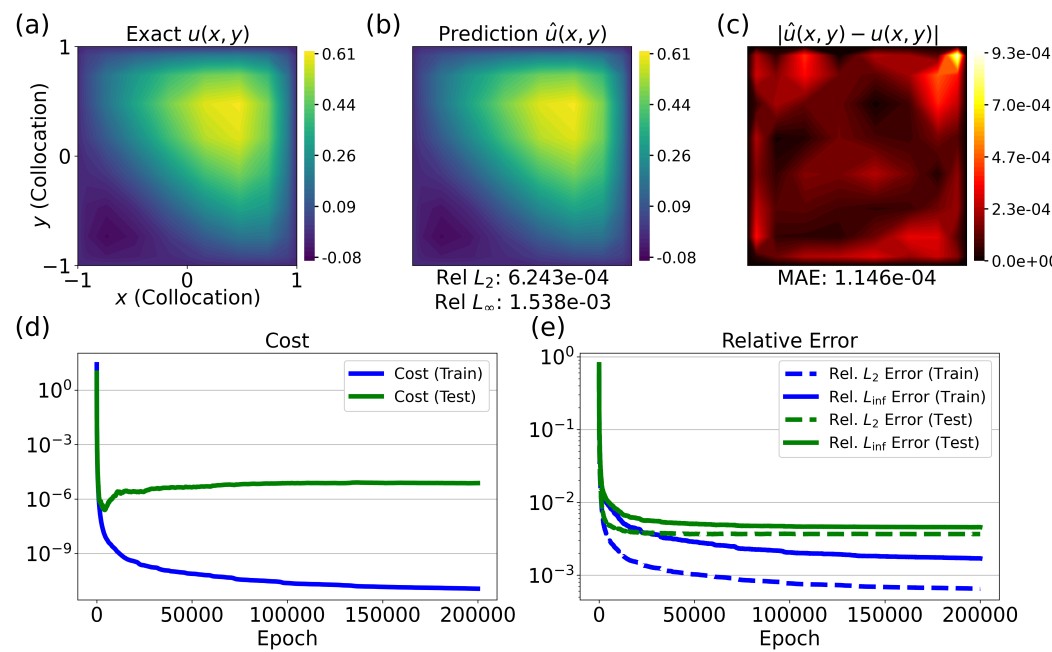

Figure 21: A representative example for the two-dimensional reaction–diffusion equation with the homogeneous Dirichlet boundary condition and a diffusion coefficient $\epsilon = 0.1$. Top: (a) The exact solution $u$, (b) The predicted solution $\hat{u}$ of NVQLS, (c) Absolute error between $\hat{u}$ and $u$. Bottom: (d) Batch-wise training and test losses, (e) Batch-wise relative $L_2$ and $L_\infty$ errors over epochs.

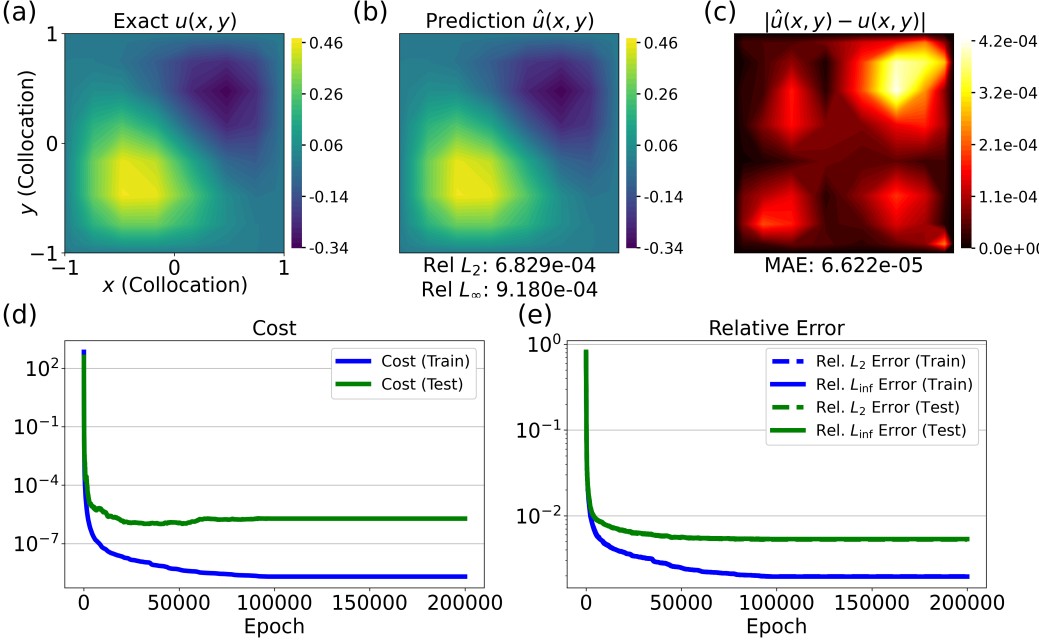

Figure 22: A representative example for the two-dimensional Helmholtz equation with the homogeneous Dirichlet boundary condition and a wave number $k^2 = 10.6$. Top: (a) the exact solution $u$, (b) the predicted solution $\hat{u}$ of NVQLS, (c) absolute error between $\hat{u}$ and $u$. Bottom: (d) batch-wise training and test losses, (e) batch-wise relative $L_2$ and $L_\infty$ errors over epochs.

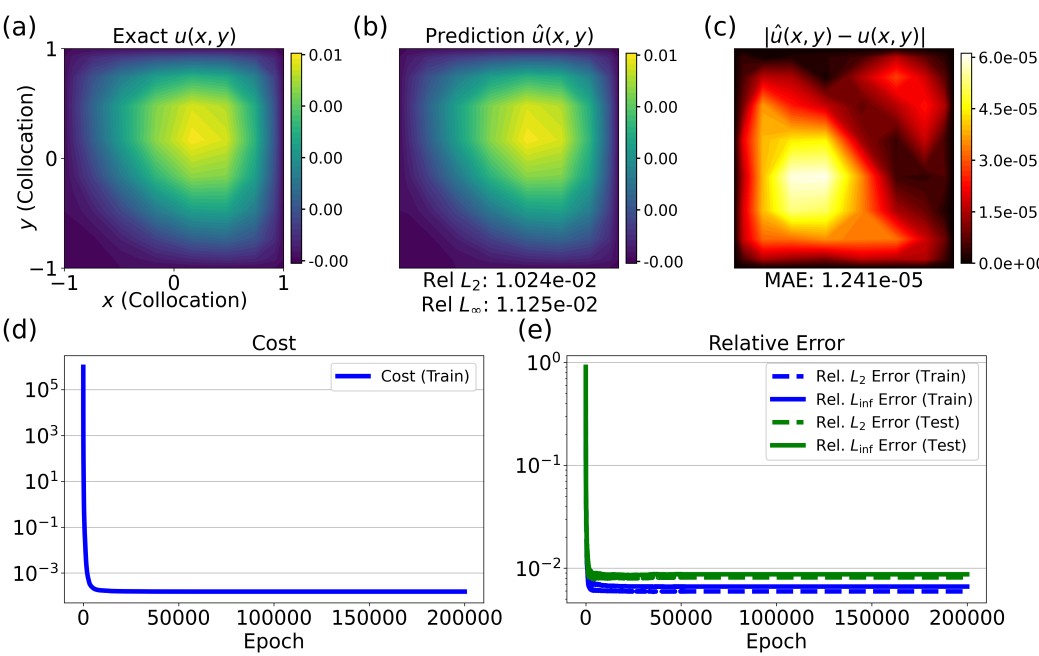

Figure 23: A representative example for the two-dimensional convection-diffusion equation with the homogeneous Dirichlet boundary condition and a wave number $\epsilon = 30.0$. Top: (a) the exact solution $u$, (b) the predicted solution $\hat{u}$ of NVQLS, (c) absolute error between $\hat{u}$ and $u$. Bottom: (d) batch-wise training and test losses, (e) batch-wise relative $L_2$ and $L_\infty$ errors over epochs.

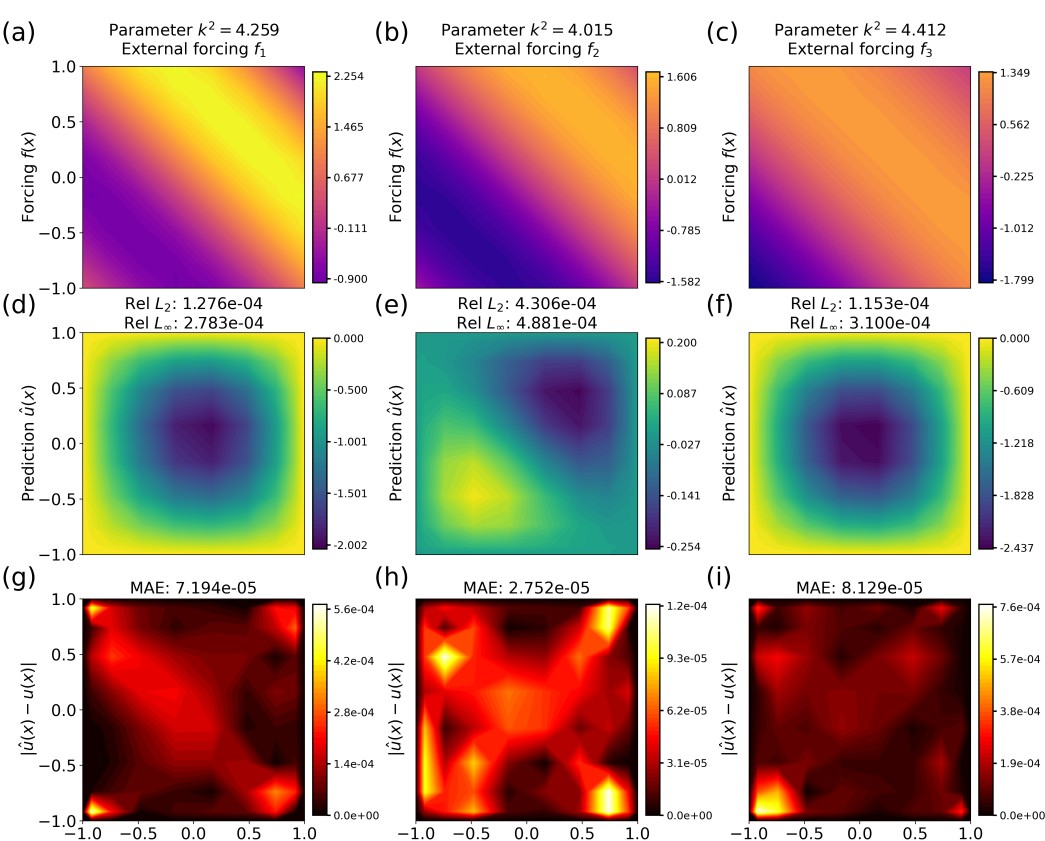

Figure 24: Numerical examples of operator learning with joint parameter and forcing inputs for the two-dimensional Helmholtz equation under Dirichlet boundary conditions. Top: input pairs for the angle network: (a) case 1: a forcing function $f_1$ and a diffusion coefficient $k^2 = 4.259$, (b) case 2: $f_2$ and $k^2 = 4.015$, (c) case 3: $f_3$ and $k^2 = 4.412$. Middle: predicted solutions $\hat{u}_i$ for each case: (d) case 1, (e) case 2, (f) case 3. Bottom: absolute error plots $|\hat{u}_i - u_i|$ on collocation points excluding the boundary points (g) case 1, (h) case 2, (i) case 3.

