# OpenReview forum: "Quantum spectral operator learning for solving partial differential equations"
_ICLR.cc/2026/Conference — Submitted to ICLR 2026_

### Official Review · Reviewer_pt9b · 2025-11-01

**Soundness:** 3
**Presentation:** 3
**Contribution:** 2
**Rating:** 2
**Confidence:** 4

**Summary:**

The author provides a hybrid quantum-classical method to solve the PDE by incorporating the classical neural network parameterised quantum states, the VQLS solver, and the improved loss function. The result and the proposed advances are very clear, with sufficient experiments.

**Strengths:**

1. The improved loss function reduced the complexity of the VQLS algorithm.
2. The parameterised model "seems" to have better representation capacity than the original parameterised state in VQLS.

**Weaknesses:**

1. The demonstration of why the neural network parameterised state is better lacks persuasion.
2. The simplification of the loss function VQLS is quite trivial. (Please correct me if I am wrong)

**Questions:**

1. What is the cost of using only the real part of equation 5 as the target?
2. Why neural network parameterised state better than the usual VQLS method? Please clarify this from a theoretical aspect.

---

> ### Author Response · Authors · 2025-11-24
>
> [Weakness 1] We appreciate the reviewer’s question, as it provides an opportunity to directly clarify the representational advantages of our NVQLS framework over the standard VQLS approach. The proposed NVQLS framework leverages a classical neural network as the angle network that maps a batch of PDE instances to quantum circuit parameters. This makes three advantages;
>
> (1) enabling multi-instance training,
>
> (2) eliminating the need for data embedding quantum circuits, and
>
> (3) reducing the overall computational cost.
>
> First, in the original VQLS formulation, the multi-instance training is not possible, since the forcing function must be encoded through a fixed, problem-dependent quantum data-embedding circuit. Because VQLS provides no mechanism to parameterize multiple forcing functions, extending it to multi-instance learning would require duplicating an entire circuit for each instance, leading to computation that scales linearly with the number of inputs and becomes impractical. On the other hand, NVQLS utilizes a classical network to embed the information of PDE instances, thereby predicting multiple solutions from a batch of PDE instances. Furthermore, this neural embedding scheme eliminates the need of quantum data-embedding circuits, yielding an additional reduction in computational cost.
>
> Next, the introduction of the angle network also reduces the computational complexity compared to the classical counterpart of NVQLS in terms of Big-O analysis. While the classical counterpart directly predicts the solution coefficients with dimensionality $K=(N-1)^d$, the angle network of NVQLS outputs the quantum circuit parameters, which scale as $O(n)=O(\log K)$ due to the use of shallow quantum circuits. This reduction in output dimensionality leads to a further decrease in computational cost, as illustrated in Table 1 of the manuscript.
>
>
>
> [Weakness 2] Thank you for giving us the opportunity to elaborate on the overlap loss function and the accompanying quantum circuit design methodology. In fact, the proposed overlap loss function
>
> (1) substantially reduces the computational cost and
>
> (2) resolves the intrinsic sign ambiguity of VQLS-type formulations.
>
> First, the proposed structure of an overlap-based loss function also leads to computational reductions through two design choices: i) an efficient measurement scheme that exploits observable commutativity, and ii) a reformulation of the loss numerator from a double-sum to a more scalable single-sum structure. Since the proposed structure of an overlap-based loss function places the measurement block at the very end of the quantum circuit, this enables simultaneous measurement of commuting terms, significantly reducing the measurement cost as shown in newly added Figure 8 and 9. Furthermore, since only the real part of the overlap is required, the double sum of the original VQLS loss collapes to a single sum, further simplifying the calculation.
>
> Next, it is worthy to note that the original VQLS constructs its loss using the fidelity, i.e., the squared magnitude of the overlap, which inherently removes the sign information of the PDE solution. This is not an artifact of implementation but a theoretical limitation of the fidelity-based objective itself, leading to a reflection of predicted solutions, as illustrated in newly added Figure 1. The overlap loss function fundamentally resolve this issue, which directly uses the real part of the quantum-state overlap and is, to our knowledge, the first formulation that preserves the sign information of quantum linear solvers based on VQLS.
>
> We appreciate the reviewers’ insightful comments. Based on their feedback, we are revising the presentation of our contributions.

---

> > ### Author Response · Authors · 2025-11-24
> >
> > [Q1] We appreciate the reviewer’s important question, which directly relates to the main contribution of our approach -- specifically, the use of the real part of Equation 5 as the target. The main improvement is not merely a simplification of the loss function, but devising new quantum circuits with significant reduction in the gate complexities and runtime for calculating such loss function. The new design is not trivial at all and our work is the first one to introduce it.
> >
> > If one uses a fidelity-based loss, it is indeed possible to adopt the circuits in Figure 9 (b-c) of the original VQLS paper. However, such a fidelity loss inevitably suffers from a sign ambiguity. In addition, the VQLS circuit requires evaluating combinatorial combinations of index pairs (e.g., $l, l'$), which leads to $O((K \log K)^2)$ computational cost, where $K = (N-1)^{d} =  2^{n}$ and $n$ denotes the number of qubits. As a result, there is no practical speedup compared to the classical ULGNet baseline.
> >
> > On the other hand, a naive implementation of our overlap-based loss would require estimating quantities of the form $\langle f|A_l|\alpha\rangle$ via a Hadamard test. This, in turn, demands controlled versions of all unitaries involved: the ansatz preparing $|\alpha\rangle$ from $|0\rangle$, the operator $A_l$, and the unitary preparing $|f\rangle$. Moreover, this procedure has to be repeated for each of the L Pauli terms, leading to deep circuits and substantial overhead.
> >
> > Our proposed method avoids these issues: we do not need to control the application of $A_l$, which directly reduces the circuit depth. Furthermore, by pushing $A_l$ to the measurement stage and treating it as part of the observable, we can group mutually commuting Pauli operators and estimate all terms in each commuting group within a single circuit. This reduces the number of distinct circuits from $L$ to the number of commuting groups, which is typically much smaller than $L$. This is precisely where the non-trivial circuit-design contribution lies, beyond a simple loss-function simplification.
> >
> >
> > [Q2] We thank the reviewer for raising this point, which allows us to clarify more directly the theoretical advantages of the neural network–parameterized state over the standard VQLS ansatz. Distinct from the standard VQLS method, which, for each fixed right-hand side (RHS) $f$ in $A\alpha = f$, optimizes variational parameters to obtain an instance-specific solution $\alpha(f)$, our NVQLS framework is trained on multiple instances $\{f_i\}$ to learn a generalized solution map $f \longmapsto \alpha(f;w)$. The neural network $g(f;w)$ outputs the coefficient vector $\alpha(f;w)$, which is then encoded into the quantum state $|\alpha(f;w)\rangle = U(g(f;w))|0\rangle$. Because our goal is to approximate this solution operator for an entire family of RHSs rather than a single instance, the target coefficients $\alpha(f;w)$ necessarily require higher expressivity than in the conventional, instance-wise VQLS setting.
> >
> >
> > To achieve this purpose, we introduce a neural network that generates optimized inputs for the quantum circuit $U$. If one attempted to perform such multi-instance $f$-learning using the standard VQLS framework, one would need to find optimized quantum parameters $\theta$ such that $|\alpha\rangle = U(f,\theta)\lvert 0\rangle$.  This requires solving for $\theta$ after embedding $f$ directly into the quantum circuit.  However, by using a neural network to produce an optimized input $g(f, w)$ for the quantum circuit $U$—rather than optimizing the circuit parameters $\theta$ after embedding $f$—we directly learn the state that enters $U$. This allows the model to explore a substantially larger and more expressive subset of the states that $U$ can generate, thereby enhancing the overall representational power of the quantum circuit.
> >
> >
> > Furthermore, by embedding $f$ into $U$ indirectly through the neural network rather than encoding it directly, we eliminate the need for a separate quantum feature map. This reduces the additional circuit components required for direct embedding and consequently leads to a shallower quantum circuit—an important advantage for NISQ-era implementations.

---

### Official Review · Reviewer_2JZZ · 2025-11-01

**Soundness:** 3
**Presentation:** 3
**Contribution:** 3
**Rating:** 4
**Confidence:** 4

**Summary:**

The authors target the problem of unsupervised spectral operator learning addressing the high training costs of the classical methods for high dimensional operators or when using higher resolutions. They propose a quantum-classical hybrid framework for this.  They argue that since using the previous methods to predict the K coefficients results in squared dependence on the complexity and there instead using the classical neural network to predict the circuit parameters and then using the newly proposed quantum circuit to predict the coefficients from those angles can then lead to K log K dependence since final prediction of the coefficients is the bottleneck step in the inference using the standard classical neural networks. The authors then demonstrate their proposed scheme on 1 and 2 D PDEs where they achieve low error rates.

**Strengths:**

The overall idea seems intuitive since I agree that this can theoretically enhance the complexity by just augmenting the final coefficienct prediction step to instead using a quantum VQLS. The novel training loss for this classical quantum setup also seems fine and the optimized number of gates also further helps in reducing the complexity. The experimental results indicate the authors are able to train this system for the standard PDE tasks.

**Weaknesses:**

Like the proposal of just augmenting the final prediction via a quantum VQLS seems not a very solid technical contribution. Furthermore even though theoretical we can argue for the complexity but I am bit unsure whether this hybrid framework can actually lead to some advantages. Also the authors didn't compare the performance with the completely classical counterpart. The authors although discussed hardware efficient training in the appendix, I am unsure how much accuracy will be hurt as compared to completely classical network and in higher dimensions how much stable would be the training of this joint learning setup since now the classical network is just predicting some angle parameters as against the final input, it might harm the training of the classical part as well. Also, I didn;t find the discussion for the impact of noise in the quantum solver to the output and training.

**Questions:**

Since now the output of the classical part is changed, can this harm its training? Also, is some completely quantum algorithm possible for this? What would be the impact of the noise due to the quantum hardware on this learning which is a practical setup.

---

> ### Author Response · Authors · 2025-11-24
>
> [Weakness: contribution] We appreciate the reviewer’s thoughtful comments and would like to clarify that our contributions extend well beyond simply augmenting the final prediction.
>
> First, we introduce the overlap loss function, which (i) resolves the intrinsic sign ambiguity of VQLS-type formulations and (ii) substantially reduces the computational cost.
>
> The original VQLS constructs its loss using the fidelity, i.e., the squared magnitude of the overlap, which inherently removes the sign information of the PDE solution. This limitation leads to the sign ambiguity illustrated in the newly added Figure 1. To address this issue, we propose the overlap loss function that uses the real part of the quantum-state overlap, thereby preserving the sign information of the solution.
>
> The introduction of the overlap loss also leads to computational reductions through two design choices: an efficient measurement scheme that exploits observable commutativity, and a reformulation of the loss numerator from a double-sum to a more scalable single-sum structure. Since the proposed structure of an overlap-based loss function places the measurement block at the very end of the quantum circuit, this enables simultaneous measurement of commuting terms, significantly reducing the measurement cost as shown in newly added Figure 8 and 9. Moreover, since only the real part of the overlap is required, the double sum of the original VQLS loss collapes to a single sum, further simplifying the calculation.
>
> Second, NVQLS employs a classical neural network as the angle network that maps a batch of PDE instances to quantum circuit parameters. This enables multi-instance training, which is not possible in the original VQLS. In the original VQLS formulation, the forcing function must be encoded through a fixed, problem-dependent quantum data-embedding circuit. Because VQLS provides no mechanism to parameterize multiple forcing functions, extending it to multi-instance learning would require duplicating an entire circuit for each instance, leading to computation that scales linearly with the number of inputs and becomes impractical. On the other hand, NVQLS utilizes a classical network to embed the information of PDE instances, thereby predicting multiple solutions from a batch of PDE instances. Furthermore, this neural embedding scheme eliminates the need of quantum data-embedding circuits, yielding an additional reduction in computational cost.
>
> The introduction of the angle network also reduces the computational complexity compared to the classical counterpart of NVQLS in terms of Big-O analysis. While the classical counterpart directly predicts the solution coefficients with dimensionality $K=(N-1)^d$, the angle network of NVQLS outputs the quantum circuit parameters, which scale as $O(n)=O(\log K)$ due to the use of shallow quantum circuits. This reduction in output dimensionality leads to a further decrease in computational cost, as illustrated in Table 1 of the manuscript.
>
> Finally, through numerical experiments, we demonstrate that NVQLS can predict solutions in both 1D and 2D settings when the forcing function is used as the data instance, achieving error metrics below 1%. Furthermore, in a more general setting where both PDE parameters and forcing functions serve as joint inputs, NVQLS continues to perform effectively in both 1D and 2D cases. These results indicate that, even when the underlying PDE operator is even nonlinear and highly complex, the proposed NVQLS framework is capable of learning this operator effectively.
>
>
> [Weakness: comparison]
> In newly added Appendix Figure 9, we include a direct comparison with the fully classical ULGNet baseline. Both models are trained on the same dataset of forcing functions generated via rotations of the Pauli-Y operator, so that each input vector consists of sine–cosine components. For the 1D Helmholtz equation, we use comparable neural-network sizes (12,688 parameters for ULGNet vs. 11,080 for NVQLS) and train on exactly the same dataset. As shown by the relative errors on the test set, NVQLS achieves better accuracy than ULGNet, demonstrating an improvement over the completely classical counterpart.
>
> [Weakness: joint input training for 2D PDE]
> We appreciate the opportunity to demonstrate the stability of joint-input training in higher dimensions. We conducted experiments with the 2D joint-input setting, and the results are shown in newly added Figure 6. Conceptually, these experiments follow the same principles as the 1D joint-input training described in the main text, with no significant modifications other than the use of a 2D matrix. These results confirm that the NVQLS framework’s joint-input training naturally and stably extends to higher dimensions. Details of the neural network and ansatz structure for these experiments will be added to the appendix.

---

> > ### Author Response · Authors · 2025-11-24
> >
> > [Q1] We understand the concern regarding potential difficulties in training due to the change in the classical network output. Importantly, the use of the classical network in NVQLS does not harm training, because it allows direct embedding of multiple PDE instances into the quantum circuit parameters while keeping the ansatz shallow.
> >
> > In contrast, in the standard VQLS framework, incorporating information about the forcing term requires adding forcing-embedding layers before a parameterized quantum circuit. NVQLS eliminates the need for such embedding layers by using a classical neural network to encode the forcing information directly into the ansatz parameters. By leveraging the expressivity of the classical network, NVQLS reduces the number of quantum layers required, enabling efficient and stable training.
> >
> >
> > [Q2] Regarding the possibility of a completely quantum algorithm, we note that implementing multiple PDE instances would require (i) an additional quantum embedding layer to encode multiple forcing terms, and (ii) a sufficiently expressive ansatz to represent several forcings simultaneously. Achieving this would typically require a significant increase in the depth of the quantum circuit, which could lead to higher computational cost and increased susceptibility to hardware noise.
> >
> > In contrast, NVQLS uses a classical neural network to embed multiple forcing terms, thereby eliminating the need for a separate forcing-embedding layer. By leveraging the expressivity of the classical network, NVQLS can effectively predict the solution coefficients using only a shallow ansatz.
> >
> > [Q3]
> > We appreciate the reviewer’s question regarding the practical use of the algorithm. Hardware noise is indeed an important issue. In this work, our focus is on proposing a hardware-efficient, circuit-parameterized training scheme. By estimating $A_l$ via direct Pauli measurements, our circuits use fewer controlled gates and result in shallower circuits, helping to mitigate the impact of hardware noise. Nevertheless, actively suppressing or correcting hardware noise itself lies beyond the scope of the present study.
> >
> > There have been several efforts to develop noise-resilient variational quantum circuits [1-3], and as a natural extension of this line of research, our future work will aim to design algorithms that are more robust to hardware noise in the NISQ era.
> >
> > References
> >
> > [1] William J Huggins, Sam McArdle, Thomas E O’ Brien, Joonho Lee, Nicholas C Rubin, Sergio Boixo, K Birgitta Whaley, Ryan Babbush, and Jarrod R McClean. Virtual distillation for quantum error mitigation. Physical Review X, 11(4):041036, 2021 DOI: https://doi.org/10.1103/PhysRevX.11.041036
> >
> > [2] Van Den Berg, Minev, and Temme] Ewout Van Den Berg, Zlatko K Minev, and Kristan Temme. Model-free readout-error mitigation for quantum expectation values. Physical Review A, 105(3):032620, 2022. DOI: https://doi.org/10.1103/PhysRevA.105.032620
> >
> > [3] Kim, Christopher J Wood, Theodore J Yoder, Seth T Merkel, Jay M Gambetta, Kristan Temme, and Abhinav Kandala. Scalable error mitigation for noisy quantum circuits produces competitive expectation values. Nature Physics, 19(5):752–759, 2023. ULR http://dx.doi.org/10.1038/s41567-022-01914-3

---

> > > ### Comment · Reviewer_2JZZ · 2025-11-26
> > >
> > > Thanks for a detailed rebuttal! However, I am not fully convinced with the use-case of the proposed algorithm with the justifications provided and the overall technical contribution. Also, the latest experiments do not strongly advocate in terms of performance since they are limited to 1/2 D and when using the quantum hardware for any experiments, will inherit noise which can harm the performance, especially in higher dimensions. Due to the lack of robustness and no major demonstration of empirical advantage in the higher dimensions and and not very convincing technical contribution, I would stick to my original rating.

---

### Official Review · Reviewer_873B · 2025-11-01

**Soundness:** 2
**Presentation:** 3
**Contribution:** 3
**Rating:** 4
**Confidence:** 4

**Summary:**

This paper proposes NVQLS, a hybrid quantum-classical framework for the unsupervised operator learning of Partial Differential Equations (PDEs). The method first discretizes the continuous PDE into a high-dimensional linear system $A\alpha = F$ via spectral methods. It then trains a classical neural network (an "Angle Network") to learn the map from the PDE instance $f$ (which defines $F$) to the parameters $\theta$ of a variational quantum circuit. This quantum circuit, a modified Variational Quantum Linear Solver (VQLS), then prepares the solution state $|\hat{\alpha}\rangle$. The authors claim two primary contributions: (1) A new "overlap" cost function (Eq. 7) that solves the standard VQLS's "sign ambiguity" problem and reduces its measurement complexity. (2) A full hybrid framework that, by using the classical network to learn a "compressed" $K \to \log K$ map, allegedly reduces the overall complexity from the classical $O(K^2)$ to $O(K \log K)$.

**Strengths:**

1. A Novel and Critical Improvement to VQLS: The paper's most significant contribution is the new "overlap cost function" (Eq. 7). This is a genuine and valuable innovation. It correctly identifies two crippling flaws in the standard VQLS (Eq. 5): its $O(L^2)$ complexity and its "sign ambiguity" (due to the fidelity loss $\propto |\cdot|^2$). The proposed loss function elegantly solves both problems simultaneously: it is linear (not quadratic), thus preserving the solution's sign, and its numerator reduces the complexity to a single summation $O(L)$. This makes VQLS a much more viable algorithm for physical problems like PDEs.
2. Intelligent Hybrid Framework Design: The proposed AI-quantum architecture is well-motivated. It correctly identifies the $O(K^2)$ bottleneck of purely classical spectral operator learning. The proposed "division of labor"—using a classical NN for the "easy" $K \to n$ (where $n=\log K$) parameter mapping and a quantum circuit for the "hard" $n \to K$ state preparation—is a theoretically sound and intelligent approach to tackling this classical scaling challenge.

**Weaknesses:**

Despite the strength of its proposed loss function, the paper's central claim of a "quantum advantage" (i.e., the $O(K \log K)$ complexity) rests on a fatal theoretical assumption that is never justified. Furthermore, the experimental validation is methodologically flawed, failing to provide convincing evidence of the model's true accuracy.
1. The "Quantum Advantage" Claim is Based on an Unjustified Expressibility Assumption: The paper's entire $O(K \log K)$ complexity argument is built on a "missing" assumption: that a shallow quantum circuit with $O(\text{poly}(n)) = O(\text{poly}(\log K))$ parameters has sufficient expressibility to approximate the $K$-dimensional solution $\alpha$ to a high degree of accuracy.
2. The Core Complexity Claim is Theoretically Unsound: This assumption is well-known to be false. The quantum variational algorithm literature has established that for an ansatz to be "universal" (i.e., able to approximate any state in the $K$-dimensional space), its depth must scale exponentially with $n$, i.e., $O(\text{poly}(K))$. A shallow $O(\text{poly}(\log K))$ ansatz can only represent a vanishingly small fraction of the solution space. The paper completely ignores this fundamental trade-off between ansatz depth (cost) and approximation error (accuracy), which invalidates its central complexity claim.
3. Critical Hyperparameters are Omitted: Compounding the theoretical flaw above, the paper is critically vague about its experimental setup. It provides extensive detail on the classical network's architecture (Table 3), but I could not find any mention of the quantum circuit's depth ($l_V$). This is the single most important hyperparameter for determining the model's expressibility and cost, and its omission is a serious flaw in reproducibility and transparency.
4. Inappropriate Plotting for True Solution: While interpolation plotting for the predicted solution is acceptable due to the resolution limitations of the model, it is problematic that the true solution also seems to use interpolated plots. This raises concerns regarding the validity of the error calculations, as it is unclear whether the true solution was computed at the same resolution as the predicted solution or at a higher resolution. The authors should clarify this issue and ensure that the true solution used in error computations is based on a numerically reliable resolution.

**Questions:**

See weaknesses.

---

> ### Author Response · Authors · 2025-11-24
>
> We appreciate the reviewer's comment. We aim to address this concern by providing both theoretical and empirical discussions on the use of shallow circuits, along with detailed explanations of our experimental methodology, to fully resolve the reviewer's objection.
>
> [Weakness 0,1,2] We sincerely thank the reviewer for raising this important theoretical point. While shallower circuits may seem less expressive, recent studies
> [1-4] show that model accuracy depends not only on expressibility but also on trainability.
>
> This can be vied as a trade-off between the size of the representable space and learnability, and overly deep circuits can hinder training due to barren plateaus. Hence, for the specific problems considered, a carefully chosen shallow circuit can provide sufficient expressibility while maintaining stable and efficient training.
>
> As illustrated in newly added Figure 8, for various ansatz structures considered, we observe that increasing the depth beyond 12 does not significantly enhance expressibility, which is quantified as the distance to the Haar distribution of random unitary operators; a smaller distance corresponds to greater expressibility.
>
>
> [Weakness 3] We sincerely thank the reviewer for raising this important point. To improve clarity, we summarize the circuit depths used in our experiments in Table 4. All circuits remain shallow, with depth growing only logarithmically in the number of qubits, and none exceeding a depth of 20. Specifically, we used depth 12 for 1D PDEs, and slightly larger depths for 2D PDEs and joint-input operator learning to ensure sufficient expressibility. These settings confirm that our experiments operate within a reasonable circuit regime and support the validity of the complexity analysis presented in the paper.
>
>
> [Weakness 4] We used numerical solutions based on a spectral method as the ground truth, and both were evaluated on the same grid when computing the errors. Classical spectral convergence results [5-7] ensure that the approximation error decays extremely fast so that, with a suitable number of basis functions, the numerical error becomes effectively close to the limits of machine precision, typically around $10^{-12} \sim 10^{-14}$.
>
> In our experiments, for the number of qubits ($n = 4, 5$), the corresponding spectral basis sizes ($N = 17, 33$) are sufficient to reach this regime. Consequently, the “true” solution used for error computation is a high-accuracy spectral solution. Any interpolation of the true solution is used only for visualization in the plots and does not affect the underlying error calculations. We clarify this in A.7 Performance Metrics to ensure the validity of the numerical reference and the error computation.
>
> References
>
> [1] Yuxuan Du, Zhuozhuo Tu, Xiao Yuan, and Dacheng Tao. Efficient measure for the expressivity of variational quantum algorithms. Phys. Rev. Lett., 128: 080506, Feb 2022a. doi: 10.1103/PhysRevLett.128.080506. URL https://link.aps.org/doi/10.1103/PhysRevLett.128.080506.
>
> [2] Han-Xiao Tao, Xin Wang, and Re-Bing Wu. On the design of expressive and trainable pulse-based quantum machine learning models. arXiv preprint arXiv:2508.05559, 2025. URL https://arxiv.org/abs/2508.05559
>
> [3] Jirawat Tangpanitanon, Supanut Thanasilp, Ninnat Dangniam, Marc-Antoine Lemonde, and Dimitris G Angelakis. Expressibility and trainability of parametrized analog quantum systems for machine learning applications. Physical Review Research, 2(4):043364, 2020. doi: https://doi.org/10.1103/PhysRevResearch.2.043364
>
> [4] Adrián Pérez-Salinas, Radoica Draškić, Jordi Tura, and Vedran Dunjko. Shallow quantum circuits for deeper problems.
> Phys. Rev. A, 108:062423, Dec 2023. doi: 10.1103/PhysRevA.108.062423. URL https://link.aps.org/doi/10.1103/PhysRevA.108.062423.
>
> [5] Jie Shen, Tao Tang, and Li-Lian Wang. Spectral Methods: Algorithms, Analysis and Applications, volume 41 of Springer Series in Computational Mathematics. Springer Berlin, Heidelberg, 1 edition, 2011. ISBN 978-3-540-71040-0. doi: 10.1007/978-3-540-71041-7. URL https://doi.org/10.1007/978-3-540-71041-7
>
> [6] Richard L. Burden and J. Douglas Faires. Numerical Analysis. The Prindle, Weber and Schmidt Series in Mathematics. PWS-Kent Publishing Company, Boston, fourth edition, 1989. URL https://www.bibsonomy.org/bibtex/2ec113bf688ad26d9712fce561c087db1/jil
>
> [7] Lloyd N. Trefethen. Spectral Methods in MATLAB. Society for Industrial and Applied Mathematics, 2000. doi: 10.1137/1.9780898719598. URL https://epubs.siam.org/doi/abs/10.1137/1.9780898719598

---

### Official Review · Reviewer_sC53 · 2025-11-01

**Soundness:** 3
**Presentation:** 2
**Contribution:** 2
**Rating:** 6
**Confidence:** 4

**Summary:**

The paper presents a novel resource-efficient quantum-classical approach called the neural variational quantum linear solver (NVQLS) that combines neural operator learning with the efficiency of quantum linear solvers. The proposed approach shows a speed and performance advantage over existing methods.

**Strengths:**

The proposed approach could benefit researchers within the scientific machine learning community (both on the classical and quantum side), particularly if interested in exploring higher resource efficiency and possible performance gains for real-life problems.

The authors have evaluated their proposed method aganist ULGNet and the exact solutions for several well-known benchmark PDEs.

**Weaknesses:**

Given that the method builds upon VQLS, a known issue faced by variational algorithms is barren plateaus that may stall opimisation. Have the authors looked at how sensitive their method is to barren plateaus, particularly with deeper quantum architectures?

Furthermore, the Pauli decomposition results in a large number of terms that increase the size of the training circuit, which can cause measurement overhead.

**Questions:**

What design and hyper-parameter optimisation approaches were used to build the classical neural network part of the model?

Why was L-BFGS chosen as the default optimiser given that gradient-based methods are more performant for neural network training?

Many of the plots present the change in the cost of training and testing with epochs may show that training could've been conducted with fewer steps.

---

> ### Author Response · Authors · 2025-11-21
> **Official Comment by Authors for Reviewer sC53**
>
> Thank you for the valuable feedback on our paper. We hope that our responses will address the reviewer’s questions effectively.
>
> [Weakness 1.]
> We thank the reviewer for raising the important issue of barren plateaus.
> To assess the expressibility the ansatz, we evaluated its distance from the Haar distribution as a standard measure of expressibility illustrated in newly added Figure 8. Our analysis shows that this distance decreases as the number of layers increases, indicating enhanced expressivity. From these observations, we first explain a depth range in which the ansatz exhibits sufficiently high expressibility. Within this expressive regime, we investigated which layer depths yield reliable optimization behavior. Some depths exhibited degraded performance, likely due to unfavorable local minima or the onset of barren plateaus. Nevertheless, we identified a subset of depths that permit stable optimization. We attribute this to the neural network, which allows the model to explore the parameter landscape more effectively than standard VQLS optimization, facilitating the discovery of regions that support stable and well-conditioned training.
>
>
> [Weakness 2]
> Thank you for raising this important question regarding the efficiency of our model. The number of Pauli measurements is indeed a key factor determining the overall computational efficiency, and our proposed framework is specifically designed to reduce this cost. As a result, we achieve a meaningful reduction in computational complexity compared with both the original VQLS framework and its classical counterparts.
>
> This reduction is empirically demonstrated in the newly added Figures 9 and Figure 10, which we will include in the revised manuscript. Figure 9 presents the Pauli decomposition size of the matrix $A$, corresponding to the numerator of the loss function for both VQLS and NVQLS. Across varying numbers of basis functions and dimensions, NVQLS consistently shows a significant reduction in measurement requirements, empirically achieving a scaling smaller than sub-$N^d$. Similarly, the computational cost of the Pauli decomposition of $A^\dagger A$, which corresponds to the denominator of the loss function, is clearly illustrated in Figure 10.
>
> Furthermore, to provide a direct comparison with classical methods, we include a Big-O complexity analysis contrasting NVQLS with ULGNet. This comparison is detailed in Section 3.2 and Table 1. While the classical counterpart has an output dimension of $O(N)$, the angle network of NVQLS produces the output whose dimension is only  $O(n)=O(\log N)$, since this output is used as quantum circuit parameters of the shallow circuit structure. This reduced output dimensionality is a primary contributor to the improved computational efficiency of our method.

---

> > ### Author Response · Authors · 2025-11-24
> >
> > [Q1] Thank you for your question regarding the design and optimization of the angle network. While full architectural details are already provided in Appendix A.3 and Table 3, we did not describe the hyper-parameter settings. Hence, we plan to include details on the heuristic, progressive hyper-parameter tuning strategy in Appendix A.3 to further support reproducibility.
> >
> > For 1D PDEs, we used standard feed-forward networks for simplicity and efficiency. For 2D PDEs, we adopted a CNN–FF hybrid to capture local spatial patterns, using GELU activation (performance was similar with ReLU) and variance-scaling initialization from JAX Flax.
> >
> > Regarding hyper-parameter optimization, we used a heuristic, progressive approach. Starting with a minimal architecture, we modified width and depth only when it improved generalization of the model. Since NVQLS relies on shallow quantum circuits, small architectures were sufficient for stable training.
> >
> > [Q2] We appreciate the reviewer’s question regarding the key details of our experimental setup. In many PDE-related and scientific machine learning tasks, L-BFGS is commonly preferred, as such problems often involve complex, stiff, or nonlinear optimization landscapes where quasi–second-order methods tend to offer improved stability and convergence. As the reviewer noted, we also experimented with gradient-based optimizers such as Adam for the one-dimensional convection–diffusion problem. However, in our experiments, L-BFGS consistently demonstrated more stable convergence behavior, and thus we adopted it as the default optimizer for the results presented in the paper.
> >
> > [Q3] Thank you very much for this helpful observation. We agree that the learning curves in Figures [16-19, 21-23] indicate that convergence is effectively reached well before the final training epoch. In our original setup, we fixed the number of epochs to 200,000 for both Adam and L-BFGS in order to use exactly the same optimization budget for all experiments and keep the comparison between optimizers methodologically fair, even though L-BFGS typically converged much earlier than Adam.
> >
> > To address your suggestion more directly, we also evaluated the model performance using the recorded metrics at epochs 50k, 100k, 150k, and 200k, and report the corresponding training and test relative errors in Tables A.3 and A.3. The resulting errors change only marginally (for example, the RD 1D with Dirichlet BC maintains a relative $L^2$ error of $1.773\times10^{-3}$ already from 50k epochs onward, while the Helmholtz 1D with Neumann BC decreases from $8.199\times10^{-3}$ at 50k epochs to $6.277 \times 10^{-3}$ at 100k epochs and remains at that value thereafter; see revised Table A.5), confirming that our conclusions do not depend on the longer training schedule. In the revised manuscript we now clarify this choice of a fixed epoch budget and explicitly note that, in practical applications, a smaller number of epochs (e.g., 100,000) is sufficient to achieve essentially the same accuracy.

---

### Author Response · Authors · 2025-12-03
**Summary of Rebuttal Discussion for New Area Chair**

Summary of Rebuttal Discussion for New Area Chair
========

During the rebuttal period, we fully addressed all questions from the reviewers and attached newly added figures and tables along with numerical results of an additional experiment.


- __Main Contributions__

  Q. Reviewers 2JZZ and pt9b expressed uncertainty regarding the authors’ contributions. Hence, we clarified our contributions and will rearrange the manuscript to assist readers in clearly understanding the contributions of the proposed framework.

  A.

  **Contribution 1.** The overlap loss function (i) resolves the sign ambiguity of VQLS, which has not been addressed in prior work, and (ii) substantially reduces computational cost.

  **Contribution 2.** This work is the first to propose the multi-instance training framework based on VQLS, which (i) uses the angle network that maps a batch of PDE instances to quantum circuit parameters and (ii) decreases the computational cost in terms of Big-O analysis due to its reduced output dimensionality.

  **Contribution 3.** We introduce the efficient joint input training framework, empirically validated through both 1D and 2D experiments.



Additional Experiment
---

- **Two-dimensional Joint Input Training (Figure 24)**

  Q. Reviewer 2JZZ questioned the training stability of the joint learning setup in higher dimensions.

  A. We conducted an additional experiment on the two-dimensional joint learning framework, and the results are shown in the newly added Figure 24, providing experimental evidence of the stability and scalability of the proposed framework.


Newly Added Figures and Tables
---


- __Sign Ambiguity (Figure 1)__

  Q. Reviewer pt9b stated that the simplification of the VQLS loss function is quite trivial.

  A. In addition to reducing computational cost, the overlap loss function resolves the sign ambiguity inherent in VQLS, which is elaborated in the newly added Figure 1: Illustration of the sign ambiguity in VQLS.


- __Ansatz Expressibility and Detailed Structure (Figure 7 & Table 4)__

  Q. Reviewer 873B was concerned about the insufficient expressibility of shallow quantum circuits to approximate the solution with high accuracy.

  A1. Highlighting that model accuracy depends not only on expressibility but also on trainability, we experimentally demonstrate the ansatz expressibility in the newly added Figure 7: Frobenius distance from Haar distribution.

  A2. Additionally, we clarified the detailed structure of the ansatz used in the experiments in the newly added Table 4: Details of the structure of the quantum circuit ansatz.


- __Cost Reduction Compared to VQLS (Figure 8 & 9)__

  Q. Reviewer sC53 was concerned about the computational overhead due to Pauli decomposition.

  A1. We visualize the reduced cost compared to VQLS in the newly added Figures 8 and 9: Empirical analysis of the number of required Pauli measurements in NVQLS.

  A2. The existing Table 1 already shows the decreased cost in terms of Big-O analysis.

- __Training Epoch (Table 5 & 6)__

  Q. Reviewer sC53 stated that the training curves suggest that training might have been completed with fewer steps.

  A. In A.3 Training Details, we now mention that a smaller number of epochs (e.g., 100,000) would be sufficient to achieve essentially the same accuracy, supported by Table 5 and 6: Training/Test relative L2 error across training epochs.

---

### Meta-Review · Area_Chair_w6Rd · 2026-01-06

**Summary:**

This paper proposes a quantum–classical hybrid framework for spectral operator learning to solve PDEs, aiming to reduce computational cost by combining neural operator learning with variational quantum linear solvers. Reviewers acknowledged that the problem is relevant and that the paper is technically detailed, with experiments on standard benchmark PDEs.

However, there were significant concerns about the overall technical contribution, the practical usefulness of the proposed approach, and the strength of the empirical evidence. I find reviewer pt9b’s review to be short and relatively uninformative, so I rely primarily on the more detailed comments from the other three reviewers. During the rebuttal period, no reviewer replied except reviewer 2JZZ. After reading the paper and the rebuttal, I find that the authors’ responses do not adequately address the core concerns about robustness, scalability, and demonstrated advantage. I therefore recommend rejection.

**Reviewer Concerns:**

Across reviews, a shared concern is that the paper does not convincingly demonstrate a clear technical or practical advantage of the proposed hybrid quantum–classical approach over existing classical methods. While the overlap loss and circuit design are interesting, reviewers questioned whether these ideas amount to a strong enough contribution beyond incremental improvements to known VQLS-based approaches.

Concerns were also raised about the empirical validation. The experiments are limited to one- and two-dimensional PDEs, and there is insufficient evidence that the method would remain robust or advantageous in higher-dimensional settings. Reviewers further noted that the paper does not convincingly address the impact of quantum hardware noise, which is likely to become more severe in higher dimensions and could significantly degrade performance in realistic settings.

Although the authors provided a detailed rebuttal and added explanations and experiments, reviewer 2JZZ explicitly followed up to state that they remained unconvinced by the justifications provided. In particular, this reviewer emphasized that the use cases are not well supported, the empirical results do not demonstrate a clear advantage in higher dimensions, and robustness concerns, especially related to noise and scalability, remain unresolved. The other reviewers did not follow up after the rebuttal. After checking the author rebuttal, I think the main concerns shared by the reviewers remain.

**Reviewer Scores:**

The reviewer scores were 6, 4, 4, and 2. During the rebuttal discussion, reviewer 2JZZ explicitly followed up and stated that they would stick to their original rating (4). The other reviewers did not follow up. As a result, no reviewer indicated an intention to change their score following the rebuttal or discussion.

---

### Decision · Program_Chairs · 2026-01-26

Reject